# DIRECT PREFERENCE OPTIMIZATION USING SPARSE FEATURE-LEVEL CONSTRAINTS

## ABSTRACT

The alignment of large language models (LLMs) with human preferences remains a key challenge. While post-training techniques like Reinforcement Learning from Human Feedback (RLHF) and Direct Preference Optimization (DPO) have achieved notable success, they often experience computational inefficiencies and training instability. In this paper, we propose **F**eature-level constrained **P**reference **O**ptimization (FPO), a novel method designed to simplify the alignment process while ensuring stability. FPO leverages pre-trained Sparse Autoencoders (SAEs) and introduces feature-level constraints, allowing for efficient, sparsity-enforced alignment. Our approach enjoys efficiency by using sparse features activated in a well-trained sparse autoencoder and the quality of sequential KL divergence by using the feature-level offline reference. Experimental results on benchmark datasets demonstrate that FPO achieves an above 5% absolute improvement in win rate with much lower computational cost compared to state-of-the-art baselines, making it a promising solution for efficient and controllable LLM alignments.

## 1 INTRODUCTION

Aligning large language models (LLMs) with human values and practical objectives is a critical challenge in AI development (Wang et al., 2023). Post-training methods, including fine-tuning (Wei et al., 2022; Chung et al., 2024) and alignment strategies (Tunstall et al., 2023), have played a significant role in refining LLM behavior. Among these, Reinforcement Learning from Human Feedback (RLHF) (Christiano et al., 2017; Ouyang et al., 2022) has emerged as a leading technique, integrating human feedback to guide models towards producing valuable and useful outputs. Despite its success, RLHF involves complex mechanisms such as reward modeling and policy gradients, which introduce significant training complexity and computational cost (Zheng et al., 2023b; Rafailov et al., 2024). To address these limitations, Direct Preference Optimization (DPO) (Rafailov et al., 2024) has been proposed as a more efficient alternative. Unlike reward-based methods such as Proximal Policy Optimization (PPO) (Schulman et al., 2017), DPO directly adjusts the model's output probabilities based on human preferences, reducing training complexity and computational cost. DPO-like approaches can offer a more stable and faster alignment process by bypassing the challenges associated with reward models and policy updates, making it a compelling solution for efficient LLM alignment since DPO uses a reference model to stabilize post-training.

Recent advancements in DPO focus on mainly two directions: efficiency *i.e.,* further simplifying the constraints of DPO, and controllability *i.e.,* keeping the balance between alignment and generation diversity. In terms of simplicity, methods like SimPO (Meng et al., 2024) and Odds Ratio Preference Optimization (ORPO) (Hong et al., 2024) eliminate the need for a reference model by using the average log probability of sequences as an implicit normalizer, thereby reducing memory usage and computational demands. However, DPO's performance is sensitive to the strength of constraints from the reference policy (Liu et al., 2024), and these reference-free alignment approaches (Hong et al., 2024; Meng et al., 2024) can *compromise control*, resulting in unstable training. In terms of controllability, Token-level Direct Preference Optimization (TDPO) (Zeng et al., 2024) introduces token-level rewards and sequential Kullback-Leibler (KL) divergence (Kullback & Leibler, 1951) to tackle issues related to linguistic coherence, diversity, and stability. However, it comes at the cost of *increased computational complexity*, introducing an additional sequential KL and depending on reference models, complicating the loss computation.

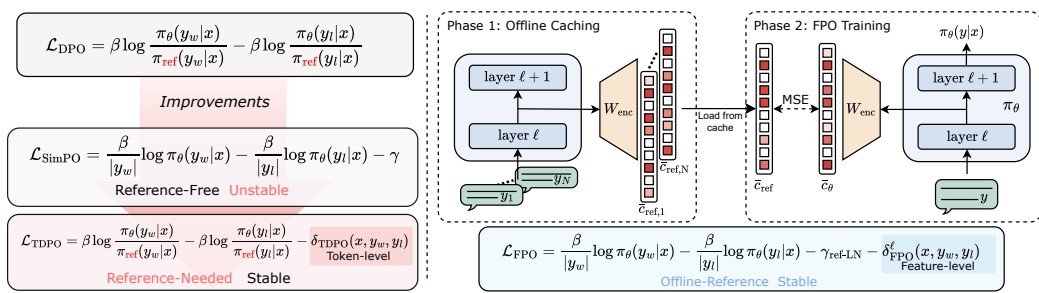

Figure 1: **Left.** The DPO objective loss function and its two main improvement directions: SimPO and TDPO. SimPO focuses on simplifying the reference model, while TDPO concentrates on controlling the alignment process to enhance generation diversity. **Right.** The pipeline of FPO consists of sparse autoencoders and the feature-level MSE constraints.

A natural hypothesis arises: "*Is there a method that can strike the right balance between efficiency and controllability?*" In response, we propose **FPO**, Feature-level Constrained Direct Preference Optimization (See Figure 1), introducing an *efficient* and *controllable* method for constraining the model at the *feature level*. Here a feature refers to a salient piece of information for the model decision (Huben et al., 2024). Intuitively, adjusting the model using feature-level preferences allows fine-grained adjustment that minimizes the side impact, by avoiding the negative influence of spurious features in course-grained control such as token level regularization (Zeng et al., 2024).

To achieve that, we derive the FPO objective by contrasting SimPO and DPO, showing the constraint term that SimPO misses. We then add such a term by introducing the **feature-level constraints** as an alternative to the costly sequential KL (Zeng et al., 2024). We use Sparse Autoencoders (SAEs) (Huben et al., 2024), which generate representations where only a few features are active, enhancing computational efficiency (See Figure 2 Right). Furthermore, regularization in the coefficient space promotes sparsity, stability, and uniqueness in the model's representations. Since SAEs produce sparse representations, only a few dozen out of 16,000 features are active at any given time (Lieberum et al., 2024). Compared to SimPO, FPO is as efficient in memory and time complexity, yet has improved controllability due to feature-level constraints; compared to constraint-based methods like TDPO, FPO matches the computational and memory efficiency of methods such as SimPO, and has potentially improved performance as feature-level control can give stronger generalization than token-level control. A contrast between FPO, DPO, SimPO and TDPO is shown in Figure 1.

Our experiments demonstrate that FPO consistently outperforms state-of-the-art methods based on different sizes of backbone LLMs, achieving up to 5% absolute improvements in win rate (See Table 2) based on AlpacaEval-2 and Arena-Hard benchmarks, up to 0.5 scores on MT-Bench and competitive output diversity. By constraining the shifts of these features during the training process, we can achieve results that meet or even exceed the effectiveness of sequential KL, at a significantly lower computational cost (17.6% reductions compared to TDPO2 as shown in Figure 4 Left). Additionally, we introduce detailed ablation studies to show that our method maintains a stable performance over different temperatures and the selection of SAE layers.

Overall, we show that FPO enjoys the efficiency of SimPO by using the offline reference control, while also the constraint quality of sequential KL by using the sparse feature-level constraints. To our knowledge, this is the first approach that integrates sparse feature-level constraints into LLM alignment. By incorporating sparse autoencoders with token-level DPO, FPO makes practically meaningful and theoretically solid improvements over existing preference optimization methods along three dimensions: simplicity of implementation, efficiency, and generation diversity.

## 2 PRELIMINARY

**Direct Preference Optimization (DPO).** DPO, derived from Reinforcement Learning from Human Feedback (RLHF), provides a direct way to align Language Models (LLMs) with human preferences without explicitly using a reward model. In practice, an LLM is prompted with a sequence $x$ (e.g., a question) to generate a corresponding sequence $y$ (e.g., an answer), where both $x$ and $y$ consist of tokens. DPO maps the reward function $r(x, y)$ to the optimal policy by minimizing the

reverse KL divergence from a reference model. This results in the following equation for the reward:

$$r(x, y) = \beta \log \frac{\pi_\theta(y|x)}{\pi_{\text{ref}}(y|x)} + \beta \log Z(x), \tag{1}$$

where $\pi_\theta(\cdot|x)$ and $\pi_{\text{ref}}(\cdot|x)$ are policy (i.e, the LLM for post-training) and reference (i.e., the base LLM) models, respectively. $\beta$ is the coefficient that governs the strength of the KL divergence penalty, $Z(x)$ is the partition function. To align with human preferences, DPO uses the Bradley-Terry (BT) model for pairwise comparisons. By incorporating the reward function into the BT model and using the negative log-likelihood, DPO computes the loss:

$$\mathcal{L}_{\text{DPO}}(\pi_\theta; \pi_{\text{ref}}) = -\mathbb{E}_{(x,y_w,y_l)\sim\mathcal{D}} \left[ \log \sigma \left( \beta \log \frac{\pi_\theta(y_w|x)}{\pi_{\text{ref}}(y_w|x)} - \beta \log \frac{\pi_\theta(y_l|x)}{\pi_{\text{ref}}(y_l|x)} \right) \right]. \tag{2}$$

Here, $\mathcal{D}$ represents the dataset with human preference pairs. $y_w$ and $y_l$ are the preferred and less preferred completions, respectively. DPO provides a direct way to align LLMs with human preferences without the explicit use of a reward model, leveraging preference comparisons.

**Simple Preference Optimization (SimPO).** SimPO simplifies preference optimization by removing the need for a reference model and aligning rewards directly with the length-normalized log-likelihood of the policy model's output. The SimPO objective can be formulated as:

$$\mathcal{L}_{\text{SimPO}}(\pi_\theta) = -\mathbb{E}_{(x,y_w,y_l)\sim\mathcal{D}} \left[ \log \sigma \left( \frac{\beta}{|y_w|} \log \pi_\theta(y_w|x) - \frac{\beta}{|y_l|} \log \pi_\theta(y_l|x) - \gamma \right) \right], \tag{3}$$

where $\gamma$ is a positive margin ensuring the reward for the preferred response exceeds that of the less preferred one by at least $\gamma$. SimPO's key innovations are (1) eliminating the reference model and (2) incorporating a *target reward margin* $\gamma$. However, while SimPO is computationally efficient, the lack of reference control (Roy et al., 2021) results in instability. As shown by Liu et al. (2024), the reference model plays a crucial role in stabilizing training and improving performance.

**Token-Level Direct Preference Optimization (TDPO).** Token-Level Direct Preference Optimization (TDPO) refines the DPO framework by operating at the token level, accounting for the sequential nature of text generation. The TDPO objective function is defined as:

$$\max_{\pi_\theta} \mathbb{E}_{x,y^{<t}\sim\mathcal{D},z\sim\pi_\theta(\cdot|[x,y^{<t}])} \left[ A_{\pi_{\text{ref}}}([x, y^{<t}], z) - \beta D_{\text{KL}}(\pi_\theta(\cdot|[x, y^{<t}])\|\pi_{\text{ref}}(\cdot|[x, y^{<t}])) \right],$$

where $A_{\pi_{\text{ref}}}([x, y^{<t}], z)$ is the token-level advantage function, and $D_{\text{KL}}(\pi_1\|\pi_2)$ denotes the KL divergence between $\pi_1$ and $\pi_2$. The first version of the loss function is given by:

$$\mathcal{L}_{\text{TDPO}_1}(\pi_\theta; \pi_{\text{ref}}) = -\mathbb{E}_{(x,y_w,y_l)\sim\mathcal{D}} \left[ \log \sigma \left( \beta \log \frac{\pi_\theta(y_w|x)}{\pi_{\text{ref}}(y_w|x)} - \beta \log \frac{\pi_\theta(y_l|x)}{\pi_{\text{ref}}(y_l|x)} - \delta_{\text{TDPO}_1}(x, y_w, y_l) \right) \right], \tag{4}$$

where $\delta_{\text{TDPO}_1}(x, y_w, y_l)$ is the difference in forward KL divergence between the preferred and less preferred completions:

$$\delta_{\text{TDPO}_1}(x, y_w, y_l) = \beta D_{\text{TDPO}_1}(x, y_l; \pi_{\text{ref}}\|\pi_\theta) - \beta D_{\text{TDPO}_1}(x, y_w; \pi_{\text{ref}}\|\pi_\theta), \tag{5}$$

and the sequential KL divergence between policy and reference output with sequence length $T$ is defined as $D_{\text{TDPO}}(x, y; \pi_{\text{ref}}\|\pi_\theta) = \sum_{t=1}^{T} D_{\text{KL}}(\pi_{\text{ref}}(\cdot|[x, y^{<t}])\|\pi_\theta(\cdot|[x, y^{<t}]))$. To further stabilize the gradient within the optimization, an improved loss function $\mathcal{L}_{\text{TDPO}_2}$ is given by replacing the regularization $\delta_{\text{TDPO}_1}$ with:

$$\delta_{\text{TDPO}_2}(x, y_w, y_l) = \alpha \left( \beta D_{\text{TDPO}}(x, y_l; \pi_{\text{ref}}\|\pi_\theta) - \text{sg}\left( \beta D_{\text{TDPO}}(x, y_w; \pi_{\text{ref}}\|\pi_\theta) \right) \right), \tag{6}$$

where $\alpha$ is an additional hyperparameter to balance between alignment and regularization, $\beta$ is the coefficient that governs the strength of the KL divergence, and sg denotes the stop-gradient operator. Unlike DPO, TDPO introduces token-level *forward KL divergence*, allowing for finer control over model alignment and diversity in generation, also introducing additional computational overhead.

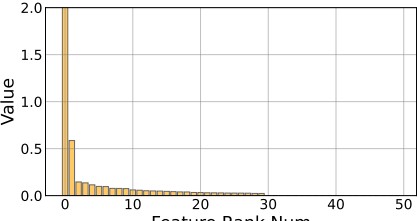

| Method | Reference | Efficiency | Constraint |
|--------|-----------|------------|------------|
| SFT | Free | **High** | Weak |
| DPO | **Offline** | **High** | Weak |
| SimPO | Free | **High** | Weak |
| TDPO | Needed | Low | **Strong / Dense** |
| FPO(Ours) | **Offline** | **High** | **Strong / Sparse** |

Figure 2: **Left.** Top-50 SAE feature activation value distribution in Gemma-2-2b. We ranked the activated feature by its activation value. The vertical axis represents the activation values, while the horizontal axis shows the rank of the maximum activation values. This plot illustrates the sparsity of SAE—out of 16,000 features, fewer than 50 have significant activation values. **Right.** Comparison of existing alignment methods on (1) if they need to load a reference model when training the policy model. (2) Memory consumption. (3) Their ability to control the generation diversity.

**Sparse Autoencoders (SAE).** SAEs provide a method for recovering monosemantic, interpretable features, enhancing the steerability of language models, where individual neurons activate in semantically diverse contexts. SAEs aim to reconstruct internal representations with sparsely activated features, disentangling the representations into interpretable components. Given the latent representation of a model $h \in \mathbb{R}^d$, its sparse activation $c \in \mathbb{R}^m$ is computed as:

$$c = \text{ReLU}(W_{\text{enc}}h + b), \quad \hat{h} = W_{\text{dec}}^T c, \tag{7}$$

where $W_{\text{enc}} \in \mathbb{R}^{m \times d}$ and $W_{\text{dec}} \in \mathbb{R}^{m \times d}$ are the learned weight matrices, $b \in \mathbb{R}^m$ is the bias vector, $m$ is the number of latent features with $m \gg d$, and $\hat{h}$ is the reconstructed input, computing loss:

$$\mathcal{L}_{\text{SAE}}(h) = \|h - \hat{h}\|^2 + \alpha\|c\|_1, \tag{8}$$

where $\alpha$ controls the sparsity of the hidden representation. The $\ell_1$-norm on $c$ enforces sparsity, ensuring only a small number of features are active at any given time (See Figure 2 Left for visualization of SAE's sparsity).

## 3  FEATURE-LEVEL DIRECT PREFERENCE OPTIMIZATION

In the right table of Figure 2, we present a comparison of FPO with other methods from three perspectives: reference model usage, efficiency, and constraint control, which is distinguished from existing methods in the following aspects:

- **Reference-free methods** such as SimPO and ORPO are memory and computation efficient. However, they struggle with instability brought by the lack of reference constraints.

- **Alignment methods with KL control on output logits**, like TDPO and KTO (Ethayarajh et al., 2024)[1], are powerful yet controllable, but their sequential KL based on output probabilities makes them costly.

- **Interpretability methods** such as SAE are widely used for interpreting the inner representations of LLMs due to their sparse and monosemantic activations Chen et al. (2017); Huben et al. (2024). However, this feature has not yet been applied in areas outside of interpretability.

**DPO with Reference-base Target Margin.** To begin, we examine the loss functions of DPO and its enhanced variants, specifically SimPO and TDPO. By comparing Equation (2) and Equation (4), we notice that TDPO and DPO share an identical implicit *reward difference* term: $\beta \log \frac{\pi_\theta(y_w|x)}{\pi_{\text{ref}}(y_w|x)} - \beta \log \frac{\pi_\theta(y_l|x)}{\pi_{\text{ref}}(y_l|x)}$. Essentially, TDPO can be viewed as an extension of DPO, where a KL constraint $\delta(x, y_w, y_l)$ is incorporated into the sigmoid function $\sigma(\cdot)$ in addition to the implicit *reward difference*. Taking a step further, we can isolate $\pi_{\text{ref}}$ from each implicit reward term:

$$\beta \log \frac{\pi_\theta(y_w|x)}{\pi_{\text{ref}}(y_w|x)} - \beta \log \frac{\pi_\theta(y_l|x)}{\pi_{\text{ref}}(y_l|x)} = \beta \log \pi_\theta(y_w|x) - \beta \log \pi_\theta(y_l|x)$$
$$- \underbrace{\beta \left(\log \pi_{\text{ref}}(y_w|x) - \log \pi_{\text{ref}}(y_l|x)\right)}_{:=\gamma_{\text{ref}}}. \tag{9}$$

---

[1]The loss function of KTO is similar to that of TDPO in terms of its use of KL divergence.

Table 1: Specific implementations of *Log Probability Difference* (LPD), *Margin*, and *Constraint* in Equation (10) for DPO, its variants SimPO and TDPO, and the proposed FPO.

| Method | LPD | Margin | Constraint | Constraint Type |
|---|---|---|---|---|
| DPO | $\beta \log \pi_\theta(y_w|x) - \beta \log \pi_\theta(y_l|x)$ | $\gamma_{\text{ref}}$ | 0 | - |
| SimPO | $\frac{\beta}{|y_w|} \log \pi_\theta(y_w|x) - \frac{\beta}{|y_l|} \log \pi_\theta(y_l|x)$ | $\gamma$ (a constant) | 0 | - |
| $\text{TDPO}_i$ | $\beta \log \pi_\theta(y_w|x) - \beta \log \pi_\theta(y_l|x)$ | $\gamma_{\text{ref}}$ | $\delta_{\text{TDPO}_i}(x, y_w, y_l)$ | KL Divergence |
| **FPO** | $\frac{\beta}{|y_w|} \log \pi_\theta(y_w|x) - \frac{\beta}{|y_l|} \log \pi_\theta(y_l|x))$ | $\gamma_{\text{ref-LN}}$ | $\delta_{\text{FPO}}(x, y_w, y_l)$ | MSE |

We can see that Equation (9) shares a similar form with the *reward difference* calculation of SimPO in Equation (3). This similarity reveals that the *reward difference* in DPO can be interpreted as a combination of log probability difference with an adaptive margin $\gamma_{\text{ref}}$ from the reference model, whereas SimPO calculates the average log probability difference with a fixed margin. Based on the above observation, we can reframe the loss function of DPO and its two variants into a unified form:

$$
\mathcal{L}_{\text{PO}}(\pi_\theta; \pi_{\text{ref}}) = -\mathbb{E}_{(x,y_w,y_l)\sim\mathcal{D}} \left[ \log \sigma \left( \underbrace{u(x, y_w, y_l)}_{\text{Log Probability Difference}} - \underbrace{\gamma(x, y_w, y_l)}_{\text{Margin}} - \underbrace{\delta(x, y_w, y_l)}_{\text{Constraint}} \right) \right].
$$
(10)

We summarize the specific implementations for DPO, SimPo and TDPO in the form of Equation (10) in Table 1. SimPO eliminates the reference model from the alignment training by using a fixed margin and omitting constraints, which reduces memory and computational costs. However, it has been criticized that completely removing reference models leads to instability (Liu et al., 2024). Our approach begins by applying the length normalization technique of SimPO to the original implicit *reward difference* of DPO:

$$
\frac{\beta}{|y_w|} \log \frac{\pi_\theta(y_w|x)}{\pi_{\text{ref}}(y_w|x)} - \frac{\beta}{|y_l|} \beta \log \frac{\pi_\theta(y_l|x)}{\pi_{\text{ref}}(y_l|x)} = \frac{\beta}{|y_w|} \log \pi_\theta(y_w|x) - \frac{\beta}{|y_l|} \log \pi_\theta(y_l|x)
$$
$$
- \underbrace{\left( \frac{\beta}{|y_w|} \log \pi_{\text{ref}}(y_w|x) - \frac{\beta}{|y_l|} \log \pi_{\text{ref}}(y_l|x) \right)}_{:=\gamma_{\text{ref-LN}}}.
$$
(11)

Equation (11) suggests using average log probability difference as the Log Probability Difference (LPD) term and introducing an adaptive margin with length normalization as the *Margin*. The length-normalized margin $\gamma_{\text{ref-LN}}$ enhances the stability by using a reference model to calculate an adaptive margin for each preference pair. We consider an offline caching technique to minimize the computational overhead introduced by the reference model.

**Feature-level Constraints.** Currently, the use of constraints $\delta(x, y_w, y_l)$ in alignment processes typically follows KL divergence-based approach shown in Equation (5) and 6. However, this method has a significant issue: for most LLMs, which generally have a very large vocabulary, where we assume the vocabulary size is $V$. For each batch with an input length of $T$, the resulting output probabilities have a size of $V \times T$. This work adopts Gemma (Lieberum et al., 2024), an advanced open-sourced LLM series, which has a massive vocabulary size of 265K. For an input length of 1024, this results in a probabilities matrix containing approximately 262M elements, which is nearly 1/10 the size of its 2B version model. Therefore, computing the KL divergence incurs considerable computational overhead to DPO-enhancing methods such as TDPO.

LLMs generate these sizable output probabilities by projecting their internal representations onto vocabulary space. In contrast to this, SAE is found to be capable of projecting these representations onto a sparse and more interpretable feature space. Motivated by the efficient nature of sparsity, we leverage the sparse feature activations from SAE to approximate the function that token-level probabilities serve in KL divergence. Specifically, for the output representation $h^{(t,\ell)}$ from layer $\ell$ of the model at position $t$, we can obtain its sparse activation $c^{(t,\ell)}$ using an SAE as described in Equation (7). Since KL divergence measures the difference between two probability distributions,

we employ MSE as the loss to measure the discrepancy between the sparse activation from the two models. To further improve efficiency, instead of calculating the sum of token-wise discrepancy like TDPO, we first perform average pooling for the sparse activation across tokens and then calculate the MSE between pooled sparse activations, which gives us a more efficient sequential discrepancy:

$$D_{\text{FPO}}^\ell (x, y; \pi_{\text{ref}} \| \pi_\theta) = \frac{1}{k} \sum_{i \in I_k} (\bar{c}_{\theta,i}^\ell - \bar{c}_{\text{ref},i}^\ell)^2, \tag{12}$$

where pooled sparse activation $\bar{c}^\ell = \sum_{t=1}^T c^{t,\ell}$, $I_k = \text{top}_k(\text{indices}(c_\theta^{(t,\ell)})) \cup \text{top}_k(\text{indices}(c_{\text{ref}}^{(t,\ell)}))$, and $\text{top}_k(\cdot)$ returns the indices of the $k$ largest elements. We focus on measuring the MSE between the largest activations to capture the discrepancy in dominant features, as these are likely to be the most influential. Echoing the strategy of TDPO, we replace $D_{\text{TDPO}}$ in $\delta_{\text{TDPO}_2}(x, y_w, y_l)$ with $D_{\text{FPO}}^\ell$ as a plug-and-play efficient approximation. This results in a feature-level constraint $\delta_{\text{FPO}}^\ell(x, y_w, y_l)$.

**Building Offline Reference Margin and Constraint.** We have justified the implementation of the key components in Equation (9), which is a SimPO-like reward difference with a reference-based adaptive margin and a feature-level constraint. At first glance, the reference model appears to be deeply involved in both the calculation of the margin and the constraint, making its complete elimination challenging. Therefore, instead of directly removing the reference model, we propose a more appropriate approach: separating the computation of the reference model from the training process by computing its output offline. Offline computation means pre-calculating and caching the results related to the reference model needed for training and then reading them during the training loop. This approach allows us to free up the reference model during alignment with only a small and acceptable I/O demand.

To explore an implementation for Equation (10) that enjoys the advantages of SimPO, such as length normalization, while ensuring stability, first, we pre-compute and store the margin $\gamma_{\text{ref-LN}}$ using the length normalization for each preference pair. Since it is scalar, it only occupies $O(N)$ space to store it, where $N$ is the number of preference pairs. Next, for the feature-level constraint, we pre-compute and store the sparse activation of each sample in the training dataset following the computation in Equation (12). Consequently, we only need to pre-compute and store one sparse activation $\bar{c}_{\text{ref}}^\ell$ for each sample, which requires $O(2 \cdot N \cdot k)$ space. This results in a significantly smaller space requirement compared to constraints used in TDPO, where the vocabulary size is $V$, for each batch with $N$ preference pairs, requiring a much larger space of $O(2 \cdot N \cdot V)$. By combining all the above results, we arrive at the loss function for FPO:

$$\mathcal{L}_{\text{FPO}}(\pi_\theta; \pi_{\text{ref}}) = -\mathbb{E}_{(x, y_w, y_l) \sim \mathcal{D}} \Big[ \log \sigma \Big( \frac{\beta}{|y_w|} \log \pi_\theta(y_w|x) - \frac{\beta}{|y_l|} \log \pi_\theta(y_l|x) \\ - \gamma_{\text{ref-LN}} - \delta_{\text{FPO}}^\ell(x, y_w, y_l) \Big) \Big]. \tag{13}$$

## 4 EXPERIMENTAL SETUP

**Model and Training Settings.** Our model selection is guided by two key principles: scalability and transparency. For scalability, we first select a series of models spanning different parameter sizes, including Gemma-2-2B and Gemma-2-9B (Team et al., 2024)[2]. This ensures that we can evaluate our approach's performance as the model parameters scale and assess its robustness across diverse model architectures. For transparency, we exclusively select foundational models, which have not undergone supervised fine-tuning (SFT) or alignment processes. We begin by fine-tuning these models using a unified conversational format provided by the Halos dataset, applying it to the Ultrachat-200K (Ding et al., 2023). Dataset for initial instruction tuning. This establishes a baseline conversational capability and ensures that all our methods are compared on a consistent SFT model. Subsequently, we employ the UltraFeedback (Cui et al., 2024). Dataset to align the SFT models using various methods. This approach maintains transparency and control throughout the process, as all data and methods are open-sourced across the experimental setup.

For the hyperparameters related to alignment methods, such as $\alpha$ and $\beta$, we initially refer to the hyperparameter settings from the corresponding papers. If these settings are explicitly provided, we

---

[2]We select Gemma-scope as it provides pre-trained SAEs (Lieberum et al., 2024) for all layers.

Table 2: Performance comparison of different methods for Gemma-2-2B and Gemma-2-9B across various benchmarks (AlpacaEval-2, Arena-Hard, and MT-Bench), compared to Supervised Fine-Tuning (SFT), DPO and variants. Length controlled Winning Rate: WR-L; Winning Rate: WR.

| | Gemma-2-2B | | | | Gemma-2-9B | | | |
|---|---|---|---|---|---|---|---|---|
| **Method** | **AlpacaEval-2** | | **Arena-Hard** | **MT-Bench** | **AlpacaEval-2** | | **Arena-Hard** | **MT-Bench** |
| FPO v.s. | WR-L(%) | WR (%) | WR (%) | △ Score | WR-L (%) | WR (%) | WR (%) | △ Score |
| SFT | 54.7 | 55.1 | 53.2 | +0.5 | 51.2 | 52.4 | 53.4 | +0.3 |
| DPO | 51.7 | 50.8 | 51.6 | +0.1 | 51.0 | 51.0 | 51.2 | +0.1 |
| TDPO-1 | 51.5 | 54.4 | 51.4 | +0.3 | 50.8 | 50.2 | 51.8 | +0.1 |
| TDPO-2 | 50.9 | 54.0 | 50.6 | +0.2 | 50.2 | 49.9 | 49.5 | 0.0 |
| SimPO | 51.1 | 52.2 | 51.4 | +0.4 | 50.2 | 51.8 | 51.0 | +0.2 |

directly adopt their configurations. For configurations that are not given, we perform a hyperparameter search to determine the optimal values. Regarding the training hyperparameters, we standardize the batch size to 32, set the learning rate to $5 \times 10^{-7}$, and use a warm-up period of 150 steps, after which the learning rate remains constant, set the epoch as 1. We employ the Adam (Kingma, 2014) and RMSProp optimizers (Graves, 2013) for Gemma-2-2B and Gemma-2-9B, respectively.

**Baseline Methods.**    Regarding our baseline comparison methods, we primarily compare three categories of approaches. The first category consists of our foundational methods, including instruction fine-tuning (SFT) and DPO itself. Here, SFT refers to the model's performance after the first-stage fine-tuning, while DPO refers to the direct application of DPO for further alignment following SFT. The second category includes methods with explicit KL control and efficient reference-free methods. We select the TDPO series *i.e.,* TDPO-1, TDPO-2 and SimPO, as they currently represent the state-of-the-art in these two classes of methods (DPO-enhancing and DPO-simplified), respectively.

**Evaluation Benchmarks.**    We evaluate our models on three widely-used open-ended instruction-following benchmarks: MT-Bench (Zheng et al., 2023a), AlpacaEval 2 (Li et al., 2023; Dubois et al., 2024), and Arena-Hard (Li et al., 2024; Chiang et al., 2024). These benchmarks are designed to test the models' conversational abilities across a broad spectrum of tasks and have gained significant adoption in the research community. AlpacaEval 2 includes 805 questions derived from five different datasets, while MT-Bench spans eight categories with a total of 80 questions. Arena-Hard, the most recent release, builds on MT-Bench by introducing 500 complex technical problem-solving queries.

We follow the standard protocols for each benchmark in evaluations, by computing the △**Score** as the margin between FPO and other methods. The metrics evaluated include Length Controlled Winning Rate (WR-L) and Winning Rate (WR) for AlpacaEval-2 and Arena-Hard, and a score from 1-10 for MT-Bench. For all methods, we use GPT-4 -Turbo (Achiam et al., 2023) as the evaluator.

For analyzing the alignment and diversity trade-off of our method, following Zeng et al. (2024), in experiments, we validate and compare FPO against several strong alignment baselines, including DPO (Rafailov et al., 2024), SimPO (Meng et al., 2024), TDPO1, and TDPO2 Zeng et al. (2024).

## 5 RESULTS AND DISCUSSIONS

**FPO Consistently Outperforms Strong Baselines on Three Benchmarks.**    We evaluate the performance differences between FPO and other methods across three key aspects: training accuracy, generation diversity, and performance on downstream tasks. In terms of downstream tasks, we assess the model's performance including the winning rate or score on the AlpacaEval2 Benchmark, Arena Hard, and MT Bench. As shown in Table 2, FPO achieves highly competitive results, with up to a 5.08% improvement in winning rate compared to other methods when testing on Gemma-2-2B. Additionally, based on Gemma-2-9B, we observe a consistent improvement in our method compared to baselines. However, the performance improvements on the 9B model introduced by FPO are limited compared to the 2B model. We argue that this is because, with the same width of the SAE, smaller models, due to their lower complexity, achieve a more thorough decomposition of features, filtering more noisy features, and leading to more accurate constraints.

Table 4: Ablation Study on SAE layer selection, hyperparameters $\alpha$ and stop-gradient operator (Grad. sg. for short). We perform experiments on Gemma-2-2b, with the 25th layer's residual SAE used to evaluate the effects of varying $\alpha$ and applying a stop-gradient. We search for the best settings considering the trade-off between Alignment (accuracy) and Diversity (entropy).

| Search Strategy: Layer Selection | | | | | | | | |
|---|---|---|---|---|---|---|---|---|
| layer $\ell$ | 7 | 7 | 13 | 13 | 19 | 19 | **25** | 25 |
| SAE type | Residual | MLP | Residual | MLP | Residual | MLP | **Residual** | MLP |
| Accuracy (%) ↑ | 57.2 | 57.4 | 59.1 | 61.3 | 59.7 | 62.4 | 63.6 | 63.4 |
| Diversity (Entropy) ↑ | 1.645 | 1.609 | 1.612 | 1.637 | 1.644 | 1.654 | 1.680 | 1.671 |
| Search Strategy: $\alpha$ Selection / Stop-Gradient | | | | | | | | |
| $\alpha$ | 0.1 | 0.5 | 1 | 2 | 0.1 | **0.5** | 1 | 2 |
| Grad. sg. | - | - | - | - | Chosen | **Chosen** | Chosen | Chosen |
| Accuracy (%) ↑ | 64.1 | 63.7 | 63.4 | 61.9 | 64.0 | 63.6 | 62.7 | 62.1 |
| Diversity (Entropy) ↑ | 1.630 | 1.642 | 1.666 | 1.643 | 1.652 | 1.680 | 1.682 | 1.679 |

## 5.1 THE TRADE-OFF BETWEEN CONTROLLABILITY AND EFFICIENCY

**Accuracy vs. Diversity.** We measure the training accuracy on the UltraFeedback dataset, which is defined as the probability that the chosen answer's token-wise probabilities exceed those of the rejected answer. Table 3 shows the model's generation diversity by measuring the entropy of the top 100 results on AlpacaEval2, where the ↑ indicates higher values are preferable. We use **bold** to show the best-performing result across all metrics, and underline to denote the second-best result. The results indicate that FPO achieved the second-highest training accuracy, only behind TDPO2, outperforms other baselines, and has the highest diversity. We also demonstrate that FPO exhibits entropy levels comparable to methods like TDPO-2, which excel in controlling output diversity, indicating the effectiveness of FPO.

Table 3: Comparison of FPO and other baseline methods in terms of the trade-off between Alignment (accuracy) and Diversity (entropy) on the UltraFeedback dataset.

| Method | Accuracy (%) ↑ | Diversity (Entropy) ↑ |
|---|---|---|
| DPO | 59.9 | 1.66 |
| TDPO-1 | 63.2 | 1.65 |
| TDPO-2 | **64.2** | **1.68** |
| SimPO | 63.4 | 1.64 |
| FPO | _64.1_ | **1.68** |

**FPO Yields Better Controllability and Efficiency Trade-off.** Using Gemma-2-2B as the base model, we first conduct dialogue fine-tuning and proceed with the testing phase. For the calculation of KL divergence, we consistently apply TDPO's sequential KL divergence method. Specifically, we compute the KL divergence of the policy model relative to the reference model for both the preferred response (*i.e.,* chosen) and the dispreferred response (*i.e.,* rejected). The results (See Table 3) indicate that, due to FPO's excellent KL control and well-designed reward structure, it achieves performance comparable to other methods while maintaining lower computational costs.

**Hardware Efficiency of FPO.** Given the efficiency of FPO compared to TDPO2, as shown in the left one in Figure 4, we consider this result to be highly competitive. The efficiency of FPOis reflected primarily in two aspects: (1) Offline Processing. FPO does not require an additional reference model to be loaded during training, but only incurs minimal I/O overhead to read pre-stored information at each step, specifically the one-dimensional tensors needed for training. This process can be efficiently handled by the dataloader. (2) Sparsity. Due to the sparse activation values in the SAE encoder, we only need to process the activated values, reducing computational overhead. To validate its efficiency, we tested the memory consumption of different methods during training. In terms of memory usage, FPO maintains nearly the same level of memory consumption as reference-free methods like SimPO. Compared to methods that introduce more computation, such as TDPO, FPO achieves approximately a 17% memory optimization.

It is important to note that, compared to reference-free methods like SimPO, FPO still requires pre-computation of the reference model's log probabilities and SAE feature activations. However, this reduces the peak computational and memory demands, making the model easier to run on smaller devices with lower costs. Considering that scaling up computational resources is generally more challenging than extending runtime, we believe this represents a reasonable trade-off between perfor-

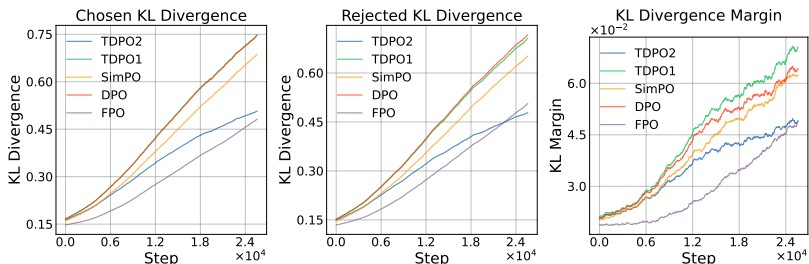

Figure 3: **Left.** KL Divergence on the preferred responses (chosen). **Center.** KL Divergence on the dispreferred responses (rejected). **Right.** KL Divergence margin *i.e.,* $|\beta D_{\text{SeqKL}}(x, y_l; \pi_{\text{ref}} \| \pi_\theta) - \beta D_{\text{SeqKL}}(x, y_w; \pi_{\text{ref}} \| \pi_\theta)|$. The KL margin is a key indicator of balancing alignment and diversity during generation. A significant margin often results in the model becoming overly focused on a narrow subset of preferred responses while suppressing responses that are deemed less aligned.

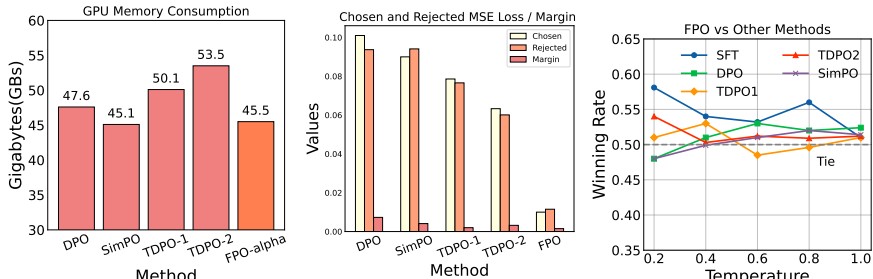

Figure 4: **Left.** GPU memory consumption on a single H100 with all methods. We average the average GPU memory in 1,000 steps at the beginning of the training. **Center.** Feature-level MSE Loss of all methods after the whole alignment process. Here margin is defined as $|D^\ell_{\text{FPO}}(x, y_l; \pi_{\text{ref}} \| \pi_\theta) - \beta D^\ell_{\text{FPO}}(x, y_w; \pi_{\text{ref}} \| \pi_\theta)|$. The close correspondence between the MSE Loss margin reduction and KL divergence margin reduction supports the validity of our approach. **Right.** Win rates of FPO v.s. other methods above the improvements based on Gemma-2-2B evaluated by GPT-4 on different sampling temperatures.

mance and cost. Additionally, it is worth noting that the hardware requirements during the caching process are significantly lower, as the model is only in inference mode.

**Consistency between MSE Loss and KL Divergence.** In TDPO and KTO, the use of KL divergence serves to constrain the margin between the model's preferred response (chosen) and dispreferred response (rejected), thereby allowing for better control over the dispreferred responses. We also evaluated the margin between chosen and rejected responses under MSE Loss across 32 response sets (see Figure 4). The results indicate a high degree of consistency between the constraints enforced by MSE Loss and those enforced by KL divergence (see Figure 3 and Figure 4). Through these constraints, the model reduces the deviation in the distribution of dispreferred responses.

## 5.2 ABLATION STUDY

To validate the insertion position of the SAE encoder and the settings of other hyperparameters, we conduct an ablation study as shown in Table 4. We train Gemma-2-2B on UltraFeedback for one epoch to evaluate the performance of different configurations. In terms of metrics, we focus on accuracy and diversity (measured by entropy) to balance alignment and diversity. Regarding the insertion position of the SAE encoder, we test the following: (1) Inserting at different layers, including shallow, middle, and deep layers. (2) Inserting the encoder after the residual stream, i.e., immediately after the residual connection to extract features, versus inserting it after the output of the MLP layer. We did not test the insertion after the attention output, as SAE is designed to capture more polysemous features in the MLP layer and the final residual output. Prior work supports this design. (3) Varying the value of $\alpha$, which affects the strength of the constraint. (4) The use of the stop-gradient operator. From Table 4, we show that inserting the encoder closer to the final output

leads to better performance. We hypothesize that this is because the layers near the final output have a more significant impact on the final result. If the encoder is inserted too early, the later layers do not receive gradients from the MSE loss, which negatively affects the model's performance. Regarding the choice of $\alpha$, we find that although a larger $\alpha$ yields stronger constraint effects while also limits the model's alignment performance. Therefore, we select $0.5$ as the optimal $\alpha$. Our tests on the stop-gradient operator demonstrate its effectiveness, which is consistent with TDPO.

**Varying Sampling Temperatures.** To investigate the performance variation of FPO under different sampling temperatures, we designed a set of temperature comparison experiments based on the ArenaHard dataset. We configured five different softmax sampling temperatures: $0.2$, $0.4$, $0.6$, $0.8$, and $1.0$. Then, for each of these temperature settings, we sampled responses from all tested methods across the first 100 questions of the ArenaHard dataset. We compared FPO's sampling results with those of other methods, using GPT-4 Turbo as the judge, and calculated a winning rate based on the win-loss results for each comparison. A winning rate greater than $50\%$ indicates that FPO achieved better alignment. As shown in Figure 4, the results show that, across multiple temperature settings, FPO outperforms other methods in at least 3-4 temperature conditions.

## 6 RELATED WORK

**Preference Optimization Methods in LLMs.** Reinforcement learning (RL) has become a popular post-training technique, enabling models to learn implicit rewards from human feedback (Ouyang et al., 2022; Dubey et al., 2024; Yang et al., 2024a). The traditional RL pipeline involves training a reward model and updating the policy model via Proximal Policy Optimization (PPO) (Schulman et al., 2017). Recent work, such as DPO (Rafailov et al., 2024), leverages the log ratio between policy and target models to directly update policies based on the reward model's objective. Extensions of DPO have introduced further refinements. KTO (Ethayarajh et al., 2024) eliminates pairwise data by modifying the value function using prospect theory, allowing training on individual sequences. Token-level DPO (Zeng et al., 2024) enforces constraints at the token level to improve generative diversity and also extends to the selection of specific tokens in pre-training (Lin et al., 2024) and post-training (Yang et al., 2024b). To reduce computational costs, ORPO (Hong et al., 2024) and SimPO (Meng et al., 2024) remove the reference model, streamlining training. Our approach similarly omits the reference model for computational efficiency but uniquely integrates feature-level constraints to achieve both high efficiency and quality in preference learning.

**Interpretating LLMs in Feature Space.** One approach to LLM alignment focuses on enhancing transparency via mechanism interpretability (Shen et al., 2023; Wu et al., 2024). A central research goal in this area is understanding how LLMs internally extract, represent, and compute human-understandable features (Rai et al., 2024; Ferrando et al., 2024). Contrary to earlier assumptions, most neurons in LLMs do not activate exclusively for specific features but form polysemantic neurons (Mu & Andreas, 2020; Gurnee et al., 2023), a phenomenon termed 'superposition' (Elhage et al., 2022), which arises from compressing numerous learnable features into a limited number of dimensions (Hänni et al., 2024). Recent work shows that sparse autoencoders (SAE) can address this by decomposing internal representations into sparse, monosemantic features, improving interpretability (Huben et al., 2024; Templeton et al., 2024; Gao et al., 2024). Due to its scalability, SAE has been used to analyze LLM monosemanticity. Yan et al. (2024) found that alignment increases monosemanticity, while Marks et al. (2023) revealed that aligned LLMs learned feedback features related to human preferences, enhancing their output alignment. However, SAE has not yet been applied to construct feature-level constraints for improving LLM alignment.

## 7 CONCLUSION

In conclusion, we proposed FPO, a novel method for efficient and stable alignment of large language models using feature-level constraints. By leveraging sparse autoencoders and pre-computed offline references, FPO reduced the computational overhead traditionally associated with alignment methods like DPO and TDPO. Our experimental results demonstrate that FPO achieved significant improvements in alignment accuracy and diversity while maintaining low resource consumption. We prove that FPO achieved improvements over current state-of-the-art methods along all three dimensions: simplicity of implementation, efficiency, and generation quality.

## REPRODUCIBLE STATEMENT

To ensure the reproducibility of our work, we have taken several steps to provide transparency in both the methodology and experimental setup. (1) All theoretical claims made in the paper are fully supported with proofs provided in the Appendix. We clearly state the assumptions and provide a detailed step-by-step explanation of our derivations to ensure clarity and completeness. (2) Details of the training process, including hyperparameters, optimizer settings, and batch sizes, are provided in Section 4 of the main text and the Appendix. We also describe the architecture of the models used, the pre-trained SAEs, and their configuration in detail. For hardware reproducibility, we have listed the GPUs used for each experiment and their respective memory consumption. (3) For reproducibility of the experiments, we utilize publicly available datasets. Detailed preprocessing steps for each dataset, including any data augmentation or filtering, are provided in the supplementary materials. Links to all datasets are included, and the steps to reproduce the exact test and training sets used in the paper are fully documented.(4) For our evaluation, we follow standard protocols and use well-established benchmarks such as AlpacaEval-2, Arena-Hard, and MT-Bench. The specific metrics and evaluation criteria are described in Section 5 of the paper, ensuring consistency and repeatability across different model sizes and configurations.

By ensuring all aspects of the work are thoroughly documented and available, we strive to make the replication of our results straightforward and accessible to the research community.

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

## A  TRAINING SETTINGS

| Model Name | Gemma-2-2b | | | | | |
| Parameters | 2B | | | | | |
| Method | SFT | DPO | TDPO-1 | TDPO-2 | SimPO | FPO |
| $\alpha$ | - | - | 0.5 | | - | 0.5 |
| $\beta$ | - | 0.1 | 0.1 | 0.1 | 2 | 0.1 |
| $\gamma$ | - | - | - | - | 0.5 | - |
| learning rate | $5 \times 10^{-7}$ | $5 \times 10^{-7}$ | $5 \times 10^{-7}$ | $5 \times 10^{-7}$ | $5 \times 10^{-7}$ | $5 \times 10^{-7}$ |
| optimizer | Adam | Adam | Adam | Adam | Adam | Adam |
| warmup steps | 150 | 150 | 150 | 150 | 150 | 150 |
| activation checkpoint | True | True | True | True | True | True |
| SAE width | None | None | None | None | None | 16k |
| GPU(s) | 4 * H100 | | | | | |
| Model Name | Gemma-2-9b | | | | | |
| Parameters | 9B | | | | | |
| Method | SFT | DPO | TDPO-1 | TDPO-2 | SimPO | FPO |
| $\alpha$ | - | - | 0.5 | | - | 0.5 |
| $\beta$ | - | 0.1 | 0.1 | 0.1 | 2 | 0.1 |
| $\gamma$ | - | - | - | - | 0.5 | - |
| learning rate | $5 \times 10^{-7}$ | $5 \times 10^{-7}$ | $5 \times 10^{-7}$ | $5 \times 10^{-7}$ | $5 \times 10^{-7}$ | $5 \times 10^{-7}$ |
| optimizer | RMSprop | RMSprop | RMSprop | RMSprop | RMSprop | RMSprop |
| warmup steps | 150 | 150 | 150 | 150 | 150 | 150 |
| activation checkpoint | True | True | True | True | True | True |
| SAE width | None | None | None | None | None | 16k |
| GPU(s) | 4 * H100 | | | | | |

Table 5: Hyperparameters for Gemma-2-2b and Gemma-2-9b.

## B BOUNDING KL DIVERGENCE WITH MSE OF SPARSE ACTIVATION

**Theorem 1.** *Let $\pi_\theta$ and $\pi_{\mathrm{ref}}$ be two models with final layer outputs $h_\theta^{t,L}, h_{\mathrm{ref}}^{t,L} \in \mathbb{R}^d$ at position $t$. Let $c_\theta^{t,L}, c_{\mathrm{ref}}^{t,L} \in \mathbb{R}^m$ be their respective sparse activation generated by a SAE. Under certain conditions, minimizing the MSE between these sparse activation values leads to a reduction in the upper bound of the KL divergence between their token probability distributions.*

*Proof.* We begin by establishing key definitions and conditions:

**Definition 1** (Sparse Activations)**.**

$$c^{t,L} = \mathrm{ReLU}(W_{\mathrm{enc}}h^{t,L} + b) \tag{14}$$

**Definition 2** (Token Logits and Probabilities)**.**

$$z^t = W_{\mathrm{out}}^T h^{t,L}, \quad p_\theta^t = \mathrm{softmax}(z^t) \tag{15}$$

**Definition 3** (KL Divergence)**.**

$$D_{\mathrm{KL}}(p_{\mathrm{ref}}^t \| p_\theta^t) = \sum_{i=1}^{V} p_{\mathrm{ref}}^t(i) \log \frac{p_{\mathrm{ref}}^t(i)}{p_\theta^t(i)} \tag{16}$$

**Condition 1** (Accurate Reconstruction)**.** *The SAE reconstructs hidden representations accurately, i.e., for some small $\epsilon > 0$:*

$$\|W_{\mathrm{dec}}^T c^{t,L} - h^{t,L}\|_2 < \epsilon \tag{17}$$

**Condition 2** (Bounded Operator Norm)**.**

$$\|K\|_2 \le M \text{ for } K = W_{\mathrm{out}}^T W_{\mathrm{dec}}^T \text{ and some } M > 0 \tag{18}$$

**Condition 3** (Small Logit Differences)**.** *The difference in logits $\Delta z^t = z_\theta^t - z_{\mathrm{ref}}^t$ is small enough for the quadratic approximation of the KL divergence to hold.*

A small $\Delta z^t$ generally exists since (1) $\Delta z^t = 0$ initially, and (2) a very small learning rate (e.g., 5e-7) is usually adopted during alignment training.

Now, we proceed with the main proof:

**Lemma 1.** *Under Condition 1, the difference in hidden representations $\Delta h^{t,L} = h^{t,L}\theta - h^{t,L}\text{ref}$ can be approximated by:*

$$\Delta h^{t,L} = h_\theta^{t,L} - h_{\text{ref}}^{t,L} \approx W_{\text{dec}}^T \Delta c^{t,L} \tag{19}$$

*where $\Delta c^{t,L} = c_\theta^{t,L} - c_{\text{ref}}^{t,L}$.*

**Lemma 2.** *The difference in logits $\Delta z^t$ is related to the difference in sparse activations $\Delta c^{t,L}$ by:*

$$\Delta z^t = K \Delta c^{t,L} \text{ where } K = W_{\text{out}}^T W_{\text{dec}}^T \tag{20}$$

**Lemma 3.** *For small $\Delta z^t$, the KL divergence can be bounded by:*

$$D_{\text{KL}}(p_{\text{ref}}^t \| p_\theta^t) \leq \frac{1}{2} \|\Delta z^t\|_2^2 \tag{21}$$

*Proof.* Using a second-order Taylor expansion and noting that the maximum eigenvalue of the Hessian of KL divergence concerning logits is $\lambda_{\max}(H) = 1$:

$$D_{\text{KL}}(p_{\text{ref}}^t \| p_\theta^t) \approx \frac{1}{2}(\Delta z^t)^T H(z_{\text{ref}}^t) \Delta z^t \tag{22}$$

$$\leq \frac{1}{2} \lambda_{\max}(H) \|\Delta z^t\|_2^2 \tag{23}$$

$$\leq \frac{1}{2} \|\Delta z^t\|_2^2 \tag{24}$$

$\square$

Combining these lemmas:

$$D_{\text{KL}}(p_{\text{ref}}^t \| p_\theta^t) \leq \frac{1}{2} \|\Delta z^t\|_2^2 \tag{25}$$

$$\leq \frac{1}{2} \|K \Delta c^{t,L}\|_2^2 \tag{26}$$

$$\leq \frac{M^2}{2} \|\Delta c^{t,L}\|_2^2 \tag{27}$$

The right-hand side is proportional to the MSE of the sparse activations:

$$\|\Delta c^{t,L}\|_2^2 = \sum_{i=1}^m (c_{\theta,i}^{t,L} - c_{\text{ref},i}^{t,L})^2 = m \cdot \text{MSE}(c_\theta^{t,L}, c_{\text{ref}}^{t,L}) \tag{28}$$

Let $I_m$ be the set of indices corresponding to the top $m$ activations. Then:

$$D_{\text{KL}}(p_{\text{ref}}^t \| p_\theta^t) \leq \frac{M^2}{2} \sum_{i \in I_m} (c_{\theta,i}^{t,L} - c_{\text{ref},i}^{t,L})^2 \tag{29}$$

$$= \frac{M^2 m}{2} \cdot \text{MSE}(c_\theta^{t,L}, c_{\text{ref}}^{t,L}) \tag{30}$$

Therefore, minimizing the MSE of sparse activation leads to minimizing an upper bound on $D_{\text{KL}}(p_{\text{ref}}^t \| p_\theta^t)$.

$\square$

# C  CONCRETE EXAMPLES OF FEATURE-LEVEL REPRESENTATIONS VS. TOKEN-LEVEL EMBEDDINGS

This section provides concrete examples and visualizations to highlight the differences between feature-level representations and token-level embeddings in our framework.

## C.1 Definitions and Intuitions

**Token-Level Embeddings:** Token-level embeddings correspond directly to the token output probabilities (*logits*) generated by a model. These embeddings are high-dimensional vectors representing each token in the model's vocabulary. For a sequence $x = [x_1, x_2, \ldots, x_T]$, the token-level embeddings at position $t$ are computed as:

$$h_t = f_{\text{token}}(x_t) \in \mathbb{R}^V,$$

where $V$ is the vocabulary size, and $f_{\text{token}}$ is the output projection from the model's hidden state.

**Feature-Level Representations:** Feature-level representations, on the other hand, are high-level abstractions derived from the model's intermediate layers. These representations capture patterns and salient features across sequences. Using a Sparse Autoencoder (SAE), the hidden state $h_t^\ell$ at layer $\ell$ can be transformed into sparse activations $c_t^\ell$, defined as:

$$c_t^\ell = \text{ReLU}(W_{\text{enc}} h_t^\ell + b),$$

where $W_{\text{enc}} \in \mathbb{R}^{m \times d}$, $b \in \mathbb{R}^m$, and $m \ll V$. This sparse activation ensures only a subset of features is active, making the representation interpretable and efficient.

## C.2 Concrete Example: A Mathematical Query

Consider the input query:

*"What is the derivative of $x^2 + 3x + 5$?"*

**Token-Level Embedding:** The token-level output probabilities for each token in the response sequence, such as *"The derivative is 2x + 3."*, involve logits for every token:

$$\text{logits} = [\log P(\text{'The'}), \log P(\text{'derivative'}), \log P(\text{'is'}), \ldots].$$

**Feature-Level Representation:** Using SAE on the 25th layer, the sparse feature representation for the same sequence might activate specific features corresponding to mathematical operations or semantic groupings:

$$c^\ell = [\text{activation}_1(\text{Polynomial}), \text{activation}_2(\text{Arithmetic}), \ldots].$$

# D  Experiments on Additional Baselines and Ablation Studies

In response to reviewer feedback, we conducted additional experiments to address their concerns and validate our methodology. These include comparisons with the SimPO+KL baseline and ablations on multi-layer sparse autoencoders (SAEs).

## D.1 Comparison with SimPO+KL

This subsection provides a direct comparison of our method against SimPO+KL. We implemented SimPO+KL following the same experimental settings in Section 4. Specifically, we tested on the Gemma-2-2B model using the AlpacaEval-2 dataset, evaluating both winning rate (WR) and length-controlled winning rate (WR-L). Results are summarized in Table 6.

**Discussion:** The results show that FPO achieves comparable or better performance than SimPO+KL in both WR and WR-L metrics. This highlights the effectiveness of feature-level constraints in maintaining both alignment quality and diversity, with a competitive computational cost.

## D.2 Ablation Study on Multi-Layer SAEs

To find out the effect of extending SAEs across multiple layers, we conducted experiments adding SAEs at different layer combinations. Table 7 presents the performance metrics when SAEs were applied to various combinations of shallow, middle, and deep layers.

Table 6: Comparison of FPO with SimPO+KL on the AlpacaEval-2 dataset. Metrics include Accuracy (%), Diversity (Entropy), WR (%), and WR-L (%).

| Method | Accuracy (%) ↑ | Diversity (Entropy) ↑ | WR (%) ↑ | WR-L (%) ↑ |
|---|---|---|---|---|
| FPO (Ours) | 64.1 | 1.68 | **51.8** | **50.2** |
| SimPO+KL | 63.6 | 1.66 | 50.8 | 50.6 |
| SimPO | 63.4 | 1.64 | 50.2 | 49.8 |
| TDPO-2 | 64.2 | **1.68** | 50.0 | 50.0 |

Table 7: Ablation study on using SAEs at multiple layers in FPO. Metrics include Accuracy (%), Diversity (Entropy), WR (%), and WR-L (%).

| SAE Layers | Accuracy (%) ↑ | Diversity (Entropy) ↑ | WR (%) ↑ | WR-L (%) ↑ |
|---|---|---|---|---|
| Single Layer (Layer 25) | **64.1** | **1.68** | **51.8** | **50.2** |
| Layers 0, 25 | 62.1 | 1.70 | 47.2 | 48.8 |
| Layers 12, 25 | 61.9 | 1.64 | 48.4 | 49.5 |
| Layers 24, 25 | 58.2 | 1.66 | 48.6 | 46.4 |
| Layers 0, 12, 25 | 51.4 | 1.66 | 47.8 | 49.8 |

**Discussion:** Results indicate that adding multiple SAE layers does not consistently improve performance and may even degrade alignment metrics (e.g., accuracy and WR). The best results were achieved with a single SAE layer (Layer 25), confirming that simplicity in feature extraction leads to more stable alignment.