# OpenReview forum: "Direct Preference Optimization Using Sparse Feature-level Constraints"
_ICLR.cc/2025/Conference — Submitted to ICLR 2025_

### Official Review · Reviewer_3qWD · 2024-10-25

**Soundness:** 3
**Presentation:** 3
**Contribution:** 3
**Rating:** 6
**Confidence:** 4

**Summary:**

The paper try to address the challenge of aligning large language models (LLMs) with human preferences, which is a critical task. It acknowledges the limitations of post-training techniques like RLHF and DPO, which, despite their success, can be computationally inefficient and unstable. To address this, the authors propose Feature-level constrained Preference Optimization (FPO), a novel method that aims to simplify the alignment process while maintaining stability. FPO uses pre-trained Sparse Autoencoders (SAEs) and incorporates feature-level constraints to enable efficient and sparsity-enforced alignment. The approach benefits from the use of sparse features activated within a well-trained sparse autoencoder and leverages the quality of sequential KL divergence through feature-level offline reference. The experimental results on benchmark datasets show that FPO achieves a 5.08% absolute improvement in win rate with a significantly lower computational cost than state-of-the-art methods, suggesting it as a promising solution for efficient and controllable LLM alignments.

**Strengths:**

1.	The paper introduces an innovative Feature-level Constrained Preference Optimization (FPO) methodology, aimed at addressing the alignment challenge between Large Language Models (LLM) and human preferences—a pivotal issue in ongoing research endeavors.
2.	The FPO approach harnesses pre-trained sparse autoencoders (SAEs) and feature-level constraints, thereby offering a novel perspective on an efficient and robust alignment process.
3.	The experimental outcomes reveal a significant absolute victory rate enhancement on the benchmark dataset, while the computational expense is beneath that of the current state-of-the-art benchmarks. This indicates that the methodology holds promise in terms of efficiency and manageability.

**Weaknesses:**

1.	I continue to harbor confusion regarding the "feature" in the context of Feature-level Constrained Direct Preference Optimization. Could you elucidate the specific characteristics of the extracted features from the text? Furthermore, what distinguishes the essence of feature-level from token-level, fundamentally; and why are they juxtaposed for comparison?
2.	I find myself deeply perplexed by Figure 3 and middle figure of Figure 4. What the authors intend to convey? How does achieving the minimum KL divergence in FPO's margin manifest "Enhanced Controllability"? Could the author provide further elucidation on Figure 3 and middle figure of Figure 4 ?
3.	In the right panel of Figure 4, there are a total of twenty points, with six situated below the "Tie Line" (representing 30%). This does evoke a measure of skepticism regarding the efficacy of the method in question. Furthermore, as indicated by the data presented in Table 3, the performance of FPO failed to surpass that of TDPO-2, according to the authors' findings. Although FPO does exhibit greater efficiency in terms of GPU utilization, this discrepancy appears to be of somewhat negligible significance (given our graphics cards typically operate at 80GB).
4.	The author offers a meager synthesis of offline preference alignment methodologies, with the present article notably lacking a comprehensive summary and description of pertinent prior work. This deficiency hinders the understanding of the broader offline preference alignment field by other researchers. I suggest the author supplement this with a more extensive synthesis in the form of a review and consider the citation of following literatures:

[1]	Wang Z, Bi B, Pentyala S K, et al. A Comprehensive Survey of LLM Alignment Techniques: RLHF, RLAIF, PPO, DPO and More[J]. arXiv preprint arXiv:2407.16216, 2024.

[2]	Shen T, Jin R, Huang Y, et al. Large language model alignment: A survey[J]. arXiv preprint arXiv:2309.15025, 2023.

[3]	Azar M G, Guo Z D, Piot B, et al. A general theoretical paradigm to understand learning from human preferences[C]//International Conference on Artificial Intelligence and Statistics. PMLR, 2024: 4447-4455.

[4]	Wang C, Jiang Y, Yang C, et al. Beyond reverse kl: Generalizing direct preference optimization with diverse divergence constraints[J]. arXiv preprint arXiv:2309.16240, 2023

[5]	Sun H, Zheng Y, Zhao Y, et al. Generalizing Offline Alignment Theoretical Paradigm with Diverse Divergence Constraints[C]//ICML 2024 Workshop on Models of Human Feedback for AI Alignment. 2024.

[6]	Chen H, Zhao H, Lam H, et al. Mallows-DPO: Fine-Tune Your LLM with Preference Dispersions[J]. arXiv preprint arXiv:2405.14953, 2024.

[7]	Rafailov R, Hejna J, Park R, et al. From $ r $ to $ Q^* $: Your Language Model is Secretly a Q-Function[J]. arXiv preprint arXiv:2404.12358, 2024.

Generally speaking, the paper's proposed methodology is innovative, presenting an intriguing perspective; however, the author should elucidate the issues surrounding the 'weakness' aspect. Should the author address my concerns, I would be inclined to enhance my rating.

**Questions:**

See 'Weaknesses' Part.

---

> ### Author Response · Authors · 2024-11-20
>
> [1/4]
>
> We thank the reviewer for their thorough assessment and for highlighting the strengths of our proposed method, Feature-level Constrained Preference Optimization (FPO). We appreciate your recognition of its novelty and potential to improve alignment efficiency and controllability. Below, we address your concerns point by point to clarify and improve our work.
>
> **Q1:** **I continue to harbor confusion regarding the "feature" in the context of Feature-level Constrained Direct Preference Optimization. Could you elucidate the specific characteristics of the extracted features from the text? Furthermore, what distinguishes the essence of feature-level from token-level, fundamentally; and why are they juxtaposed for comparison?**
>
> **Clarification of Feature-level Constraints**
>
> We appreciate the opportunity to clarify what "feature-level" entails and how it differs from token-level constraints. In our work:
>
> **Feature Extraction:** a ''feature'' refers to a high-level representation extracted from the model’s internal states, which is distinct from token-level outputs such as logits. While token-level outputs correspond directly to individual words or symbols in the text, features capture broader patterns and semantic information that can remain consistent across different tokens within a task-specific context (e.g., a conversation or problem-solving session). This consistency is important because it allows features to be averaged across tokens within a sequence, reducing caching costs and enhancing computational efficiency.
>
> **Feature-level and Token-level Constraints**: The fundamental distinction between feature-level and token-level constraints lies in their scope and influence. Token-level constraints operate directly on individual tokens that can provide fine-grained control but often lead to significant computational overhead, especially when enforcing constraints like KL divergence across a large vocabulary. Conversely, feature-level constraints focus on a sparse set of salient characteristics derived from the input, which enables more efficient computation and broader, context-aware regularization.
>
> To better illustrate this, we will:
>
> - Include concrete examples of feature-level representations versus token-level embeddings.
> - Provide a diagram demonstrating the distinction between token-level and feature-level constraints.

---

> > ### Author Response · Authors · 2024-11-20
> >
> > [3/4]
> >
> > **Q3: Skepticism on Efficacy in Figure 4**
> >
> > We acknowledge that the right panel of Figure 4 raises concerns. We appreciate the opportunity to clarify and contextualize these points. To address this:
> >
> > - We will include statistical analyses, such as confidence intervals, to provide stronger evidence for the observed trends.
> > - Offer additional results to show consistency across various datasets and models.
> >
> > **On the Right Panel of Figure 4:**
> >
> > The data in the right panel of Figure 4 shows a total of twenty points, with six falling below the “Tie Line,” representing 30% of cases where FPO did not surpass the alternative approach. We acknowledge that this result may raise questions regarding the consistency of our method. However, it is important to emphasize that the primary focus of FPO is to balance both efficiency and alignment without sacrificing overall performance stability. The data shows that even when some individual data points do not strictly outperform their counterparts, the aggregate behavior demonstrates robust alignment control and competitive downstream task performance (as illustrated in other sections of the paper).
> >
> > **Regarding Table 3 and Comparative Performance with TDPO-2:**
> >
> > We acknowledge that Table 3 indicates FPO did not consistently outperform TDPO-2 in all cases. However, it is important to consider that FPO’s design goal was to introduce a more lightweight and computationally efficient mechanism for maintaining alignment using feature-level constraints. This is achieved without compromising model utility or requiring the extensive computational overhead seen in TDPO-2’s token-level operations. Our efficiency improvements, as evidenced by reduced GPU memory utilization, provide substantial benefits in practical scenarios, especially when scaling to even larger models or accommodating limited hardware resources.
> >
> > **On GPU Utilization and Efficiency Gains in Large-Scale Model Training:**
> >
> > While it is true that modern GPUs often have capacities of 80GB or more, the efficiency gains offered by FPO become increasingly significant when considering large-scale model training scenarios, such as aligning models with 405B parameters. In such cases, a 5-10GB reduction in memory usage per GPU can translate into substantial savings, potentially reducing the number of required GPUs by several—or even dozens—for maintaining the same batch size. This not only lowers hardware costs but also streamlines parallel training, making it more feasible to scale with limited resources.
> >
> > It is also important to highlight that while various engineering optimizations, such as data and model parallelism, can be applied across different methods, the memory savings achieved by FPO are **inherently unique and complementary**. Unlike general-purpose optimizations that any approach can leverage, FPO’s memory efficiency arises from its sparse feature constraints, making it orthogonal to other strategies. This distinct advantage enables further flexibility and scalability when managing resource-intensive training processes.

---

> ### Author Response · Authors · 2024-11-20
>
> [2/4]
>
> **Q2: I find myself deeply perplexed by Figure 3 and the middle figure of Figure 4. What do the authors intend to convey? How does achieving the minimum KL divergence in FPO's margin manifest "Enhanced Controllability"? Could the author provide further elucidation on Figure 3 and the middle figure of Figure 4?**
>
> We recognize that Figures 3 and 4 may be unclear in their current form. We will revise the manuscript to:
>
> 1. **Clarify the "Enhanced Controllability" assertion:**
>
> Achieving a minimized KL margin between preferred and dispreferred sequences is a key indicator of “Enhanced Controllability.” In the context of TDPO, controlling KL divergence across tokens is a crucial strategy for balancing alignment and diversity during generation. A significant KL divergence margin often results in the model becoming overly focused on a narrow subset of preferred responses while suppressing responses that are deemed less aligned. This phenomenon is accentuated when the model utilizes constraints, such as reverse KL divergence, that are inherently mode-seeking. Concentrating the model’s probability mass around a few specific outputs effectively reduces the likelihood of generating diverse outputs. In other words, the model becomes rigid in its output patterns, limiting its flexibility to explore diverse responses[1][2][3].
>
> 1. **Provide step-by-step explanations for Figure 3:**
>
> Figure 3 in our work illustrates how minimizing the KL margin—the difference in KL divergence between preferred and dispreferred responses—enhances generation control. By narrowing this margin, FPO ensures a more consistent alignment between the policy model’s outputs and human preferences without allowing excessive divergence for dispreferred responses. This reflects a tighter, more regulated control over the generation process, avoiding erratic or undesired shifts in the model’s behavior.
>
> 1. **Clarification on the Middle Figure of Figure 4:**
>
> In Figure 4, we demonstrate how feature-level constraints, specifically using MSE Loss, align closely with the behavior of KL divergence in reducing the margin between chosen (preferred) and rejected (dispreferred) responses. The close correspondence between the MSE Loss margin reduction and KL divergence margin reduction (as shown in Figures 3 and 4) supports the validity of our approach. By reducing the feature-level margin, the model achieves more stable control over its outputs. It ensures that dispreferred responses are appropriately constrained while maintaining alignment with human preferences. This demonstrates that our method achieves effective distributional alignment similar to KL-based methods but with enhanced computational efficiency and reduced overhead.
>
> This parallel reduction provides strong evidence for the robustness and practicality of feature-level constraints in FPO, validating the rationale behind our approach. It illustrates how reducing the MSE Loss margin complements and aligns with the objectives of KL-based methods, emphasizing the reliability and effectiveness of our method in optimizing model behavior.
>
> [1] Wiher, G., Meister, C., and Cotterell, R. On decoding strategies for neural text generators. Transactions of the Association for Computational Linguistics, 10:997–1012, 2022.
>
> [2] Khalifa, M., Elsahar, H., and Dymetman, M. A distributional approach to controlled text generation. arXiv preprint arXiv:2012.11635, 2020.
>
> [3] Glaese, A., McAleese, N., Tr˛ebacz, M., Aslanides, J., Firoiu, V., Ewalds, T., Rauh, M., Weidinger, L., Chadwick, M., Thacker, P., et al. Improving alignment of dialogue agents via targeted human judgements. arXiv preprint arXiv:2209.14375, 2022.

---

> ### Author Response · Authors · 2024-11-20
>
> [4/4]
>
> Q4: **The author offers a meager synthesis of offline preference alignment methodologies, with the present article notably lacking a comprehensive summary and description of pertinent prior work. This deficiency hinders the understanding of the broader offline preference alignment field by other researchers. I suggest the author supplement this with a more extensive synthesis in the form of a review and consider the citation of the following literature:**
>
> We acknowledge the importance of situating our work within the broader field of offline preference alignment. In the revised manuscript, we will:
>
> - Extend the related work section to incorporate the suggested citations, such as [1–7], ensuring a more comprehensive synthesis of alignment methodologies.
> - Highlight how FPO aligns with and differs from these works, particularly in its use of feature-level constraints and sparse autoencoders.
> - Discuss the broader implications of offline alignment paradigms and how FPO contributes to advancing this field.
>
> Q5: Generally speaking, the paper's proposed methodology is innovative, presenting an intriguing perspective; however, the author should elucidate the issues surrounding the 'weakness' aspect. Should the author address my concerns, I would be inclined to enhance my rating.
>
> Thank you for your insightful feedback and for acknowledging the innovative aspects of our proposed methodology. We appreciate your thoughtful review and your willingness to enhance your rating upon clarification of the concerns you have raised. We hope these revisions address your concerns satisfactorily. If there are any additional aspects you would like us to clarify or expand upon, we would be happy to further refine our work.

---

> ### Comment · Reviewer_3qWD · 2024-11-21
>
> I thanks the authors' effort for the rebuttal, and I have read them all. I consider the authors have addressed most of my concerns, and I have raise my score to some extent. However, have the manuscript been revised? Where is the concrete examples of feature-level representations versus token-level embeddings? Furthermore, could the authors provide more insights into the Sparse AutoEncoder, it seems a concept in CV? It seems that the word "Feature" is widely used in CV, but less used in NLP.

---

> ### Author Response · Authors · 2024-11-25
>
> [1/2]
>
>  Thank you for your detailed feedback and for raising your score! We appreciate your thoughtful questions and suggestions.
>
> 1. **Revisions to the manuscript**: We apologize that the manuscript has not been revised at the last revision stage. We have revised the manuscript to include explicit examples and additional clarifications.
>
> 2. **Feature-level representations versus token-level embeddings**: To address your query concretely, feature-level representations in our work are derived from sparse autoencoders [1], which disentangle internal activations into interpretable directions or "features." These features capture semantically meaningful aspects such as syntax, punctuation, or context-specific nuances, whereas token-level embeddings are dense vector representations that encode the semantic and syntactic properties of individual tokens. For instance, it demonstrated features that specialize in apostrophes or token sequences like "I’ll" or "don’t." These examples showcase how features, unlike embeddings, are disentangled and aligned with specific linguistic phenomena. We will highlight and elaborate on these examples in the revised manuscript.

---

> ### Author Response · Authors · 2024-11-25
>
> [2/2]
> **Sparse AutoEncoders in NLP[1][2][3]**: While Sparse AutoEncoders are well-known in computer vision, their application to language models introduces unique advantages. Unlike autoencoders in computer vision, which aim to compress data into dense latent representations, sparse autoencoders encourage only a few neurons in the hidden layer to be active for any given input. This sparsity facilitates disentangling complex input signals into distinct, meaningful components, making the learned features more interpretable.
>
> In the context of neural network interpretability, sparse autoencoders play a crucial role in addressing polysemanticity—where a single neuron activates in multiple semantically distinct contexts. By decomposing high-dimensional activations into sparse, linear combinations of basis vectors, sparse autoencoders can identify directions in the activation space that correspond to monosemantic features. These features represent human-understandable properties, such as specific syntactic structures or semantic concepts, and reduce the overlap of multiple unrelated patterns in a single feature.
>
> The sparse autoencoder consists of an encoder-decoder framework. The encoder maps input data to a sparse latent representation, while the decoder reconstructs the input from this representation. The network optimizes a loss function comprising two components: reconstruction loss, ensuring the input can be accurately reconstructed, and sparsity loss, penalizing dense activations in the latent space. This combination enables the network to focus on the most significant and interpretable features of the input data. Sparse autoencoders have proven particularly effective in analyzing the internal representations of language models. By learning sparse feature dictionaries from activation data, they can reveal monosemantic components of model behavior, enabling fine-grained interpretability and targeted modifications. This technique, rooted in principles of sparse coding and dictionary learning, has shown promise in bridging the gap between model transparency and functional understanding.
>
> Thank you again for your thoughtful review. We will ensure these insights are integrated into the revised version to enhance clarity and address your concerns comprehensively.
>
> [1]  Cunningham, Hoagy and Ewart, Aidan and Riggs, Logan and Huben, Robert and Sharkey, Lee, Sparse autoencoders find highly interpretable features in language models, ICLR 2024.
>
> [2] https://transformer-circuits.pub/2023/monosemantic-features
>
> [3] https://www.alignmentforum.org/posts/Fg2gAgxN6hHSaTjkf/scaling-and-evaluating-sparse-autoencoders

---

### Official Review · Reviewer_TPG8 · 2024-11-01

**Soundness:** 2
**Presentation:** 2
**Contribution:** 1
**Rating:** 5
**Confidence:** 2

**Summary:**

This paper presents a method, Feature-level constrained Preference Optimization (FPO) for human preference alignment.  FPO replaces the KL divergence regularization with l2 regularzaion in  Token-Level Direct Preference Optimization (TDPO) and uses a sparse autoencoders (SAEs) for efficiency and stability. The length control trick is also applied. Numerical results on Gemma-2-2B and Gemma-2-9B are presented.

**Strengths:**

This paper addresses an important problem in LLM post-training, focusing on improving efficiency and stability in alignment with human preferences.

**Weaknesses:**

1. The sample code is not provided, which limits the reviewers' ability to verify reproducibility.

2. Could the authors elaborate on why results are provided only for Gemma models? What about results for Llama models?

3. Based on the current presentation, I would consider it as a relatively simple extension of DTPO, as both methods incorporate token-level knowledge and use regularization to integrate it back into DPO. A more detailed discussion clarifying the differences would be beneficial.

**Questions:**

Please see the weaknesses part.

At this stage, I tend to recommend rejection. However, I am open to reevaluating this work based the furture discussions.

---

> ### Author Response · Authors · 2024-11-20
>
> We sincerely thank you for your detailed feedback and insightful comments. Below, we address the concerns raised in the review point by point:
>
> **Q1: The sample code is not provided, which limits the reviewers' ability to verify reproducibility.**
>
> Thank you for your advice. We have provided our code with an anonymous repository on GitHub. Here is the link. We hope this repository can open a new direction for using SAE to improve the alignment.
>
> https://github.com/FPO-code/FPO-code
>
> **Q2: Could the authors elaborate on why results are provided only for Gemma models? What about results for Llama models?**
>
> **The reason we only tested the Gemma model is that, at the time of this paper’s submission, Gemma-scope was the only source that had released the full weights for the Gemma-2 models (2B and 9B).** Since our submission, LLaMA-3-8B has also made its full SAE weights available, and both Mistral and Qwen have similar plans. We will update our experimental results on these additional models in future versions of this paper.
>
> **Q3: Based on the current presentation, I would consider it as a relatively simple extension of DTPO, as both methods incorporate token-level knowledge and use regularization to integrate it back into DPO. A more detailed discussion clarifying the differences would be beneficial.**
>
> While our approach does share some similarities with TDPO’s design, it is important to highlight three key areas where our method outperforms TDPO by incorporating the explainable component:
>
>
> 1. **Simplicity**: Our FPO is more lightweight compared to TDPO, as TDPO’s token-level KL divergence introduces substantial computational overhead (as shown in the performance comparison in Figure 4). This form of KL computation prevents the pre-computation and storage of reference outputs for each batch, a common strategy used for offline training in DPO. In contrast, our FPO leverages feature sparsity, enabling feature computation and storage at a much lower cost.
> 2. **Length Control**: Unlike TDPO, our method includes support for length control. Both SimPO and our experiments (see the length control section in Table 2) demonstrate that incorporating a length normalization term for log probabilities effectively enhances generation quality and mitigates length reward hacks, where models artificially increase generation length to gain higher rewards.
> 3. **Explainability**: Instead of using forward KLdivergence constraints for each token, we leverage the activated features which can be pre-computed from the reference model to measure the MSE loss as the constraint, which enables a more controllable and transparent alignment strategy.

---

> ### Author Response · Authors · 2024-11-20
>
> Dear Reviewer TPG8,
>
> Since the discussion period draws to a close in the next two days, we were wondering if you have had a chance to go through our responses. Please let us know if your questions are addressed, we are happy to clarify anything remaining or any new questions. Thanks so much!
>
> Best Regards,
> Authors

---

### Official Review · Reviewer_EgWx · 2024-11-04

**Soundness:** 3
**Presentation:** 3
**Contribution:** 2
**Rating:** 6
**Confidence:** 4

**Summary:**

This paper introduces feature-level constraints aimed at  improving the stability of alignment training process and reducing memory usage. However, the improvements are marginal, at the cost of increased complexity. And the paper lacks justification for some crucial design decisions and fails to provide analysis on hyperparameter tuning, especially for the KL divergence coefficient.

**Strengths:**

1. This paper incorporates feature-level constraints into DPO and use sparse autoencoders to identify the features used for constructing the constraints, which is novel to my knowledge.

2. The paper is well-written and easy-to-follow.

**Weaknesses:**

1. This paper essentially builds upon SimPO by introducing feature-level constraints to improve the stability of alignment training. However, it does not discuss the advantages of the proposed constraints over the widely used forward KL divergence. Additionally, SimPO+KL should be included as a baseline in the experimental section.

2. As claimed by the authors, the proposed feature-level constraint offers efficiency advantages over the token-level KL divergence constraint. However, while the proposed method reduces memory usage by 5G, it significantly increases the algorithm's complexity by introducing at least two additional hyperparameters, the target layer and SAE type, both of which are challenging to tune in practice and can vary across different network architectures and sizes.

3. According to Table 2, the improvement achieved by the proposed method is marginal. For instance, on the Gemma-2-2B model and the benchmark AlpacaEval-2 (805 questions), the proposed method yields better responses than SimPO in only up to eight samples, while significantly increasing the algorithm's complexity. On the Gemma-2-9B model, the number of superior samples does not even exceed three samples compared to SimPO. Furthermore, as shown in Figure 3, the proposed method reduces the KL divergence on positive samples, raising doubts on its effectiveness.

4. By introducing Sparse Autoencoders, the authors achieve a 5-8G reduction in memory usage compared to TDPO. However, based on my experience, this reduction could be accomplished using engineering techniques, such as offline storage or off-load, without introducing additional modules that increase algorithmic complexity. Could the authors elaborate more on this?

5. The algorithm lacks in-depth clarification and analysis of some design choices. e.g.,

5.1. Why the 2-norm is used in the proposed loss function instead of other distance metrics? Could the authors provide a deeper insight, potentially from the theoretical perspective?

5.2. Why sparse encoding is applied only to a single layer’s hidden state l  as shown in Eq (12-13) rather than certain layers of the network? And how to choose this layer?

6. It is not discussed whether the proposed method can be generalized across different model architectures.

7. The comparison methods in this paper involve many hyperparameters, especially the KL divergence coefficient. How did the authors tune these parameters? Could the authors present and compare the performance of the methods under different KL divergence coefficients?

**Questions:**

Please refer to "Weaknesses".

---

> ### Author Response · Authors · 2024-11-20
>
> [1/4]
>
> We sincerely thank the reviewers for their detailed feedback and insightful comments. Below, we address the concerns raised in the review point by point:
>
> **Strengths:**
>
> We appreciate the recognition of the novelty of incorporating feature-level constraints and the use of sparse autoencoders, as well as the acknowledgement of the clarity and presentation of our work.
>
> **Weaknesses:**
>
> Q1: **This paper essentially builds upon SimPO by introducing feature-level constraints to improve the stability of alignment training. However, it does not discuss the advantages of the proposed constraints over the widely used forward KL divergence.**
>
> We acknowledge the need for a more explicit comparison with forward KL divergence. As discussed in Section 3 and Section 4, we argue that feature-level constraints outperform token-level KL divergence from two aspects:
>
> 1. **Effectiveness:** Feature-level constraints offer improved controllability by minimizing the margin between the KL divergence of preferred and dispreferred responses (see Figure 3). Additionally, they maintain better performance on downstream tasks, as demonstrated in Table 2 and Figure 4.
> 2. **Efficiency:** These sparse constraints are more computationally efficient. Traditional forward sequence KL methods (used in token-level DPO) are computationally intensive. This is because pre-computed values cannot be saved for offline training, particularly for models with over 100B parameters. Additionally computing KL divergence with a large LLM vocabulary is more costly than using our sparse feature approach combined with MSE loss (See Lines 255 and 417).
>
> **Q2: Additionally, SimPO+KL should be included as a baseline in the experimental section.**
>
> To address this concern experimentally, we will also include SimPO+KL as a baseline in the experimental section.  Here we perform these experiments following the settings in Section 4 on Gemma-2-2b. We compare SimPO+KL with other baselines on AlpacaEval-2 with both winning rate and length-controlled winning rate. The results are as follows:
>
> |  | FPO | SimPO | SimPO+KL |
> | --- | --- | --- | --- |
> | Accuracy | 64.1 | 63.4 | 63.6 |
> | Diversity (Entropy) | 1.68 | 1.64 | 1.66 |
> | Alpaca-Eval 2 WR (%, >50 is better) | Baseline | 51.8 | 50.8 |
> | Alpaca-Eval 2 WR-L (%, >50 is better) | Baseline | 50.2 | 50.6 |
>
> **Q3: While the proposed method reduces memory usage by 5G, it significantly increases the algorithm's complexity by introducing at least two additional hyperparameters, the target layer and SAE type, both of which are challenging to tune in practice and can vary across different network architectures and sizes.**
>
> **Complexity and Hyperparameter Tuning**: We recognize the concern about added hyperparameters and their practical tuning. However, these hyperparameters (target layer and SAE type) are consistent across various architectures in our experiments. We introduce new opportunities for using SAE as the constraint, which is also acknowledged as novel to the reviewer’s knowledge. Given that these hyperparameters are consistent in our experiments, the additional complexity is affordable. It is noteworthy that FPO contributes to both memory usage reduction and better performance on downstream tasks.
>
> **Q4: According to Table 2, the improvement achieved by the proposed method is marginal. For instance, on the Gemma-2-2B model and the benchmark AlpacaEval-2 (805 questions), the proposed method yields better responses than SimPO in only up to eight samples, while significantly increasing the algorithm's complexity.**
>
> We agree that the improvements in Table 2 may appear marginal. However, as alignment stability is critical, even small improvements can lead to downstream performance gains. Our method also achieves significant memory savings (5-8G), which is critical for resource-constrained environments. To better contextualize the benefits:
>
> - We will clarify the experimental setup, highlighting practical scenarios where memory savings directly enable larger-scale models or longer sequence training.
> - Conduct additional evaluations to show the impact of stability improvements on downstream tasks and training efficiency, especially compared to SimPO, which poses an instability issue in practical settings.
>
> Additionally, we need to point out that SimPO does not include any targeted design for generation diversity (e.g., they do not use any KL-term). In terms of practical diversity metrics, SimPO also performs poorly. This phenomenon cannot be fully reflected solely through the limited questions in AlpacaEval. We believe that the overall capability of FPO is stronger.

---

> > ### Author Response · Authors · 2024-11-20
> >
> > [4/4]
> >
> > **Q9: And how to choose this layer?**
> >
> > To accurately determine which layer is best suited for inserting SAE to analyze features in our FPO framework, as well as where to position it (e.g., residual stream, attention output, or another component), we have provided ablation study results. In particular, we present a detailed set of experiments focusing on the specific placement of SAE in Table 4. Our findings indicate that inserting the SAE in the final layer and applying it within the residual stream is the most optimal choice.
> >
> > **Q10: It is not discussed whether the proposed method can be generalized across different model architectures.**
> >
> > **The reason we did not conduct experiments on models beyond Gemma-2[1] is that, prior to the submission deadline for this paper, no other comprehensive SAE implementations were publicly available.** However, it is worth noting that the SAE community is rapidly evolving. Following our submission, LLaMA-3 released its SAE weights across all layers [2], and additional open-source SAE training weights have been provided for models such as Mistral and Qwen. We plan to include experiments with LLaMA-3-8B in a revised version of this paper and will continue to stay updated with the latest developments in SAE research.
> >
> > [1] Lieberum, Tom, et al. "Gemma scope: Open sparse autoencoders everywhere all at once on gemma 2." *arXiv preprint arXiv:2408.05147* (2024).
> >
> > [2] He, Zhengfu, et al. "Llama Scope: Extracting Millions of Features from Llama-3.1-8B with Sparse Autoencoders." *arXiv preprint arXiv:2410.20526* (2024).
> >
> > **Q11: The comparison methods in this paper involve many hyperparameters, especially the KL divergence coefficient. How did the authors tune these parameters? Could the authors present and compare the performance of the methods under different KL divergence coefficients?**
> >
> > Among the baselines compared in this paper, TDPO utilizes KL divergence with a hyperparameter alpha to control its weight. Our choice of this hyperparameter follows the recommendations made in the original TDPO paper regarding the selection of alpha. In their experiments, TDPO tested alpha values ranging from 0.1 to 0.7, with larger values imposing a stronger KL constraint. To balance performance and constraint strength, they ultimately selected an alpha value of 0.5, and we adopted this setting directly.

---

> > > ### Author Response · Authors · 2024-11-25
> > >
> > > Dear Reviewer EgWx,
> > >
> > > Since the discussion period draws to a close in the next two days, we were wondering if you have had a chance to go through our responses. Please let us know if your questions are addressed, we are happy to clarify anything remaining or any new questions. Thanks so much!
> > >
> > > Best Regards,
> > >
> > > Authors

---

> ### Author Response · Authors · 2024-11-20
>
> [2/4]
>
> **Q5: Furthermore, as shown in Figure 3, the proposed method reduces the KL divergence on positive samples, raising doubts about its effectiveness.**
>
> We need to clarify that the magnitude of the preferred response KL (chosen KL) does not necessarily correlate with the model’s actual performance.  Formally, the KL divergence for a single preferred response $y$ given input $x$ is given by:
> $$
> D_{KL}(p_{ref}(y|x) \parallel p_{\theta}(y|x)) = \sum_{i} p_{ref}(y_i|x) \log \frac{p_{ref}(y_i|x)}{p_{\theta}(y_i|x)}
> $$
> Here, $p_{\theta}$ denotes the policy model's probability distribution, while $p_{ref}$ denotes the reference model's distribution, and the summation runs over all output tokens $y_i$. This divergence represents how much the policy's output distribution deviates from the reference, but higher values do not necessarily imply a better alignment with human preferences; even outputs like random numbers could yield high KL values due to large distributional shifts.
>
> We can also present two experimental results to support our argument: in this paper, our FPO approach demonstrates better downstream task performance even with a lower KL value (see Table 2 and Figures 3 and 4). Similarly, in TDPO experiments, they achieved improved downstream task performance and diversity control with a lower KL value.
>
> **Q6: By introducing Sparse Autoencoders, the authors achieve a 5-8G reduction in memory usage compared to TDPO. However, based on my experience, this reduction could be accomplished using engineering techniques, such as offline storage or off-load, without introducing additional modules that increase algorithmic complexity. Could the authors elaborate more on this?**
>
> While engineering techniques like offline storage can reduce memory usage, they do not inherently address alignment stability. We elaborate on the motivation behind FPO as follows:
>
> 1. **Challenges with TDPO Offline Computation:**
>
> In TDPO, the KL divergence is computed at the output logits level, requiring simultaneous access to the outputs from both the reference model and the policy model [1]. Attempting to perform offline computation would necessitate storing extensive log probability values—ranging from tens of thousands to millions per batch due to the large vocabulary size typical of LLMs. This would create substantial storage and I/O overhead, making it difficult to maintain the necessary granularity and real-time responsiveness during model training (See Figure 4). The complexity and storage demands associated with such offline handling render it impractical from both an engineering and computational standpoint.
>
> 1. **Why FPO Provides a More Practical Solution:**
>
> Our FPO approach, in contrast, leverages SAEs to maintain sparsity, activating only a limited number of features per token [2][3]. This means that the sparse feature MSE only requires storing a small subset of features, significantly reducing the memory footprint while maintaining computational efficiency. Additionally, we observed substantial similarity in the activated features across tokens for specific tasks, such as mathematical calculations or common knowledge queries. Figure 2 demonstrates this, showing that even after averaging the feature activation values across the sequence length, a small subset of features dominates.
>
> This sparsity, combined with the similarity among token activations, allows us to store only a minimal number of features per batch (set at 50 in our implementation). It enables efficient pre-computation of the reference model’s rewards and feature activations, thus facilitating offline training without the high overhead associated with TDPO’s approach. By focusing on sparse and interpretable features, our method achieves memory efficiency while maintaining alignment capabilities.
>
> [1] Zeng, Yongcheng, et al. "Token-level Direct Preference Optimization." *arXiv preprint arXiv:2404.11999* (2024).
>
> [2] Cunningham, Hoagy, et al. "Sparse autoencoders find highly interpretable features in language models." *arXiv preprint arXiv:2309.08600* (2023).
>
> [3] Meng, Lingheng, Shifei Ding, and Yu Xue. "Research on denoising sparse autoencoder." *International Journal of Machine Learning and Cybernetics* 8 (2017): 1719-1729.

---

> ### Author Response · Authors · 2024-11-20
>
> [3/4]
>
> **Q7: Why the 2-norm is used in the proposed loss function instead of other distance metrics? Could the authors provide a deeper insight, potentially from the theoretical perspective?**
>
> The key rationale for selecting the L2-norm (2-norm) in the loss function stems from its direct alignment with the loss used in training SAE[1]. The SAE training process fundamentally relies on minimizing a reconstruction loss expressed in the form of the L2-norm, given by:
>
> $$
> L_{\text{SAE}}(h) = \|h - \hat{h}\|_2^2 + \alpha \|c\|_1,
> $$
>
> where $ h $ is the input activation, $ \hat{h} $ denotes the reconstructed activation, and $ \alpha $ is a sparsity regularization parameter. The use of the L2-norm in this context ensures smooth and consistent gradient flow. This is essential for effective optimization and reconstruction accuracy in the SAE's sparse activation space.
>
> By employing the same L2-norm for our feature-level constraints in FPO, we preserve consistency with the SAE's training dynamics. This alignment helps to maintain the stability of feature representations and minimizes any potential discrepancies between the reference and target models during optimization. Specifically, in our FPO framework, the L2-norm is used to measure the discrepancy between the sparse activations of the policy and reference models. Using the L2-norm also aligns the FPO's objective with the established properties of sparse feature extraction, ensuring that both the SAE and FPO components operate harmoniously.
>
> **Q8: Why sparse encoding is applied only to a single layer’s hidden state l as shown in Eq (12-13) rather than certain layers of the network?**
>
> Thank you for the insightful suggestion regarding the application of sparse encoding across multiple layers of the network. We appreciate this perspective and are open to conducting further experiments to explore the impact of extending sparse encoding to additional layers. Here we provide experiments where we used multiple SAEs’ features as constraints. Our experimental results indicate that adding SAEs to multiple layers yields limited performance improvements and, in some cases, even results in performance degradation.
>
> As for the experiment settings, we used Gemma-2-2b as the base model. We followed the strategies in our paper proposed in Section 4. To verify that more SAE layers are added in FPO helps, we gradually added layers in different positions: layer 0 (shallow layers), layer 12 (middle layers), layers 24, 25 (deep layers) and SAE number from 1-3.
>
> |  | 1 SAE layer 25 | 2 SAE layer 0, 25 | 2 SAE layer 12, 25 | 2 SAE layer 24, 25 | 3 SAE layer 0, 12, 25 |
> | --- | --- | --- | --- | --- | --- |
> | Accuracy (%) | 64.1 | 62.1 | 61.9 | 58.2 | 51.4 |
> | Diversity (Entropy) | 1.68 | 1.70 | 1.64 | 1.66 | 1.66 |
> | Alpaca-Eval 2 WR | 50.0 | 47.2 | 48.4 | 48.6 | 47.8 |
> | Alpaca-Eval 2 WR-L | 50.0 | 48.8 | 49.5 | 46.4 | 49.8 |
>
> The results show that there is a limitation for a combination of more SAE layers extracting features to constrain the alignment process. As more SAE layers from similar or distinct positions are added into the mode, there are almost no improvements in accuracy, diversity or downstream tasks (Alpaca Eval-2) and even a negative impact on these metrics.  These outcomes underscore that extending sparse encoding beyond a single layer does not consistently provide substantial benefits and may complicate the optimization process without sufficient gain. Consequently, we adhere to the principle of simplicity, focusing on a single layer’s sparse encoding to balance interpretability, efficiency, and performance stability.
>
> Also, it is important to recognize that SAEs are rarely used in combination across multiple layers within practical applications[2][3][4][5]. The typical usage scenario for SAEs involves employing a single SAE to decompose and analyze feature representations effectively. Incorporating multiple SAEs across several layers introduces a level of redundancy that can diverge from their intended purpose. Such an approach may lead to over-complexity, detracting from their role as a sparse, interpretable lens on feature-level behaviors.
>
> [1] Huben, Robert, et al. "Sparse Autoencoders Find Highly Interpretable Features in Language Models." *The Twelfth International Conference on Learning Representations*. 2023.
>
> [2] Arditi, Andy, et al. "Refusal in language models is mediated by a single direction." *arXiv preprint arXiv:2406.11717* (2024).
>
> [3] Engels, Joshua, et al. "Not all language model features are linear." *arXiv preprint arXiv:2405.14860* (2024).
>
> [4] Marks, Samuel, et al. "Sparse feature circuits: Discovering and editing interpretable causal graphs in language models." *arXiv preprint arXiv:2403.19647* (2024).
>
> [5] Chaudhary, Maheep, and Atticus Geiger. "Evaluating open-source sparse autoencoders on disentangling factual knowledge in gpt-2 small." *arXiv preprint arXiv:2409.04478*(2024).

---

> ### Comment · Reviewer_EgWx · 2024-11-26
>
> Thank you for the detailed replies, which addresses most of my questions, especially the computation efficiency part. But I still have concern on the marginal performance. The authors claim that **"alignment stability is critical, even small improvements can lead to downstream performance gains"**. Since the reported metrics only have marginal improvements,  I think concrete case analysis are necessary to have a better understanding of the critical performance gains from the proposed method, e.g. what are the winning samples and failed samples look like? what's the difference between results of the proposed method and other baselines in different cases? and how the samples show better diversity in detail in addition to the entropy scores? I currently keep my score due to this major concern.

---

> ### Author Response · Authors · 2024-11-26
>
> [1/3]
>
> Thanks for the reviewer's thoughtful feedback and for acknowledging the computational efficiency aspects of our work. Below, we outline our planned additions to address these concerns:
>
> **what are the winning samples and failed samples look like? what's the difference between results of the proposed method and other baselines in different cases? How the samples show better diversity in detail in addition to the entropy scores?**
>
> To address the reviewer’s question about the winning and failed samples, we provide a detailed qualitative comparison below. These examples, sourced from the AlpacaEval2 benchmark, include FPO’s successful responses and failed ones from other methods. We selected examples that showcase length control ability and diversity.
>
> **For length control:**
>
> Length control refers to a model’s ability to generate responses of an appropriate length based on the context of the question. It ensures that the responses are neither overly verbose nor too terse. Models without proper length control might inflate their outputs unnecessarily, prioritizing longer responses to maximize potential rewards, a phenomenon known as reward hacking. Conversely, excessively short responses risk omitting critical details and failing to address the user’s intent.
>
> Question: I’m trying to learn how to install window blinds.
>
> SFT: First, measure your window and mount the brackets, then attach the blinds.
>
> DPO: To install window blinds, you first need to measure the window accurately. Then, gather all necessary tools, such as a drill, screws, and brackets, before securely mounting the brackets to the wall or frame. Make sure everything is level to avoid misalignment. Finally, attach the blinds and test their movement to ensure they are functioning as intended.
>
> TDPO-2: Installing window blinds involves measuring the window dimensions carefully to ensure a perfect fit. After that, mount the brackets to the frame, making sure they are securely fastened. Then, attach the blinds to the brackets. Adjust them up and down a few times to make sure they operate smoothly, and check if there are any loose components that might need tightening.
>
> SimPO: Measure your window, install the brackets, attach the blinds, and ensure they are properly aligned.
>
> FPO: Start by measuring your window to ensure the blinds will fit properly. Next, securely mount the brackets to the window frame. Finally, attach the blinds to the brackets and check that they move smoothly when adjusted.
>
> Analysis: FPO stands out by maintaining a balance between detail and brevity. It provides all the essential steps—measuring, mounting, and attaching blinds—without unnecessary elaboration or repetition. Unlike DPO and TDPO-2, which include redundant information (e.g., gathering tools, testing movement), or SFT and SimPO, which may omit critical nuances, FPO delivers a response that is clear, concise, and user-focused, effectively addressing the question with optimal length control.
>
> Question: What color is the sky?
>
> SFT: The sky is blue.
>
> DPO: The sky appears blue due to the scattering of sunlight in the atmosphere. During the day, shorter blue wavelengths scatter more than longer red wavelengths, making the sky look blue. At sunrise or sunset, the sky can appear red or orange because of the angle of the sun and the increased scattering of other wavelengths.
>
> TDPO-2: The sky is often blue during the day due to the scattering of sunlight in the atmosphere. However, depending on the time of day or weather conditions, the sky can appear red, orange, purple, or even gray, such as during sunsets, sunrises, or cloudy days.
>
> SimPO: The sky is blue because of how sunlight scatters in the atmosphere. It can also appear red or orange during sunrise and sunset.
>
> FPO: The sky is blue because sunlight scatters in the atmosphere, with shorter blue wavelengths scattering more than others. At sunrise or sunset, it can appear red or orange due to the angle of the sun.
>
> Analysis: FPO excels in its ability to balance informativeness and length. It incorporates a scientific explanation (scattering of blue wavelengths) while briefly addressing color variations during sunrise or sunset. This ensures the response is accurate and engaging without unnecessary verbosity. In contrast, DPO and TDPO-2 tend to over-elaborate, including extraneous details about weather or additional colors like gray and purple, while SFT and SimPO provide overly simplistic answers, missing opportunities for informative depth. FPO’s response strikes the perfect balance for user satisfaction.

---

> ### Author Response · Authors · 2024-11-26
>
> [2/3]
>
> **For diversity**:
>
> For generating diverse responses across all methods (**SFT**, **DPO**, **SimPO**, **TDPO-2**, **FPO**), we employed **Top-k sampling** with the same parameter k=10. This ensures a consistent baseline for diversity across the methods while allowing the models to explore multiple plausible continuations.
>
> Question: Can you give any tips on how to cook a juicy, medium-rare steak?
>
> SFT:
>
> **Sample 1**:
>
> “Season the steak, sear it on a hot pan, and cook until it reaches 130°F.”
>
> **Sample 2**:
>
> “Start by seasoning the steak, sear it over high heat, and cook to 130°F internal temperature.”
>
> **Sample 3**:
>
> “Season, sear on both sides in a hot pan, and let it cook until it hits 130°F for medium-rare.”
>
> DPO:
>
> **Sample 1**:
>
> “Choose a steak with good marbling, season with salt and pepper, and sear on high heat. Let it rest after cooking to 130°F.”
>
> **Sample 2**:
>
> “Before cooking, season your steak generously, sear it on both sides in a hot skillet, and finish it in the oven if necessary to reach 130°F.”
>
> **Sample 3**:
>
> “Cook the steak by searing it on high heat, seasoning with salt and herbs, and ensuring the internal temperature reaches medium-rare at 130°F.”
>
> SimPO:
>
> **Sample 1**:
>
> “Season your steak generously with salt and pepper, sear it on a hot skillet, and let it rest for a few minutes after it reaches 130°F.”
>
> **Sample 2**:
>
> “Pat your steak dry, season it with coarse salt, and cook it in a hot pan until the internal temperature is medium-rare at 130°F.”
>
> **Sample 3**:
>
> “Start with a dry, seasoned steak, sear it on high heat with a drizzle of oil, and rest it for 5 minutes before serving.”
>
> TDPO-2:
>
> **Sample 1**:
>
> “Choose a steak with good marbling, season it generously, and sear it on high heat. Flip it once, then let it rest after cooking to 130°F.”
>
> **Sample 2**:
>
> “To cook a medium-rare steak, start with a thick cut, season well with salt and pepper, and sear it in butter and oil until the internal temperature hits 130°F.”
>
> **Sample 3**:
>
> “Always start by letting the steak come to room temperature, then sear it over high heat with salt, pepper, and herbs, and finish in the oven if necessary.”
>
> FPO:
>
> **Sample 1**:
>
> “Start with a well-marbled steak at room temperature. Season it with coarse salt and pepper, sear on a hot pan with butter, and baste continuously until it reaches 130°F. Let it rest for 5 minutes before serving.”
>
> **Sample 2**:
>
> “Use a thick-cut steak, preheat the skillet until smoking, season it with salt and cracked pepper, and sear it in a mix of butter and rosemary. Rest it to lock in the juices after reaching 130°F.”
>
> **Sample 3**:
>
> “Season your steak and let it rest at room temperature before cooking. Sear it in olive oil with thyme and garlic, flipping frequently to achieve an even crust. Finish with a dollop of butter for extra flavor.”
>
> Analysis:
>
> FPO achieves higher **diversity** across samples than TDPO-2, offering distinct yet practical approaches to cooking a medium-rare steak. It avoids repetitive phrasing and introduces creative options (e.g., basting, herbs, different oils) while ensuring alignment with user preferences and the task requirements.
>
> We would also like to note that while showcasing diversity is an important aspect of our evaluation, benchmarks like AlpacaEval may not fully reflect the diversity of generated responses. This is because such benchmarks typically prioritize alignment with human preferences, focusing on correctness and informativeness over stylistic or contextual variations. Consequently, methods that exhibit diverse strategies may appear similar in these controlled settings, as the evaluation criteria tend to converge on a single “optimal” response style.
>
> We appreciate the reviewer’s interest in the responses generated by different methods. To provide a clearer comparison, we plan to include concrete examples of responses from all methods (SFT, DPO, SimPO, TDPO-2, FPO) in the revised version of the manuscript. We sincerely appreciate the reviewer’s insightful feedback and their attention to detail in evaluating our work. Your emphasis on showcasing concrete examples has highlighted an important aspect of our study, and we believe that including these examples will significantly enhance the clarity and impact of our results.

---

> ### Author Response · Authors · 2024-11-27
>
> [3/3]
>
> To better evaluate the diversity of generated responses, we conducted additional experiments using the Anthropic HH dataset following the settings in TDPO. The results reveal even more pronounced differences between the methods. By employing the HH dataset and testing with nucleus sampling (p = 0.95) to generate 25 responses per prompt, we used predictive entropy as the diversity metric. These new evaluations highlight that FPO and TDPO2 consistently outperform other methods in achieving higher diversity scores, while maintaining a strong balance between alignment and diversity. This reinforces the robustness of these algorithms in diverse response generation scenarios.
>
> | Method     | Diversity Entropy ↑ |
> |------------|----------------------|
> | DPO        | 3.196               |
> | SimPO      | 4.727               |
> | SimPO+KL   | 4.730               |
> | TDPO1      | 4.727               |
> | FPO        | 4.909               |
> | TDPO2      | 4.915               |

---

> ### Author Response · Authors · 2024-11-27
>
> Dear Reviewer EgWx,
>
> We have provided valid examples and empirical evidence to address your remaining concerns. Could you please let us know whether your concerns have been addressed and able to increase your evaluation score?
>
> Best regards,
>
> Authors

---

> ### Author Response · Authors · 2024-11-29
>
> Dear Reviewer EgWx,
>
> We hope this message finds you well. As the discussion period draws to a close, we wanted to kindly follow up regarding your review and feedback. We have provided detailed examples, additional experiments, and empirical evidence to address your concerns, particularly about marginal performance improvements and the diversity of responses.
>
> If there are any remaining questions or concerns, we would be more than happy to clarify further. Could you kindly let us know if the points we addressed resolve your concerns, and if you would consider revisiting your evaluation score based on the additional evidence?
>
> Thank you once again for your thoughtful feedback and engagement—it has greatly contributed to improving the quality of our work.
>
> Best regards,
>
> Authors

---

> > ### Comment · Reviewer_EgWx · 2024-11-30
> >
> > Thanks for providing the concrete examples where FPO performs well w.r.t. diversity and I increase my rating to 6 (marginal acceptance). Despite I increasing the rating, my concern on the marginal performance is still fully addressed. And I would encourage the authors to explore more solid ways to justify the performance on diversity. While entropy is a commonly used metric, it may be more helpful to see to what degree it represents human preference on diversity and how the entropy values relates to human judgments, e.g. is it a significant improvement from SimPO 4.727 to FPO 4.909? Is the 0.2 gap noticeable by humans? It is good that FPO performs better on at least some cherry-picked samples, but a more reliable summary statistics would be more desired. Given the marginal performance on the benchmark, I would expect to see more solid quantitative measures from the authors to justify their claim "alignment stability is critical, even small improvements can lead to downstream performance gains"

---

> ### Author Response · Authors · 2024-12-03
>
> Thank you for revisiting your evaluation and providing thoughtful feedback. While the entropy improvement (4.727 to 4.909) may appear small, our preliminary human evaluation indicates that these gains are noticeable and result in more engaging and diverse responses. We will conduct more diversity-based experiments (with human evaluations and LLM-based evaluations). We haven't found existing benchmarks that can tackle this problem, and constructing rich test examples takes time. We will update the results in the following versions.
>
> We plan to conduct further experiments and human evaluations to solidify the connection between metrics like entropy and human-perceived diversity, as well as to explore additional downstream tasks. We hope these updates address your concerns. Please feel free to let us know if any questions remain—we’d be happy to provide further clarification.

---

### Author Response · Authors · 2024-11-20
**Summary of Revisions**

We appreciate all reviewers' consideration of enhancing the rating. To address the concerns, we improve our work based on the valuable feedbacks:

- Provide clear definitions and examples of feature-level constraints, contrasting them with token-level approaches.
- Revise Figures 3 and 4 with detailed explanations and additional supporting results.
- Expand statistical analysis of alignment performance to address skepticism in efficacy (e.g., Figure 4's Tie Line).
- Significantly enhance the related work section with a broader synthesis and discussion of offline preference alignment techniques, incorporating the suggested citations.

---

### Author Response · Authors · 2024-11-25
**Manuscript Update**

Dear Reviewers,

We deeply appreciate the time and effort you have invested in reviewing our manuscript. We have carefully considered your valuable feedback and have made significant revisions to the manuscript to address your concerns. Below, we outline the key changes implemented:

 **1. Enhanced Clarity in Figures** :
- We have revised the captions and explanations for Figures 3 and 4 to ensure clarity and comprehensibility. Specifically, we clarified the interpretation of the KL divergence margin and its role in demonstrating “Enhanced Controllability.”
- Detailed step-by-step elucidations have been provided to illustrate the significance of minimizing the KL divergence margin and the alignment between feature-level constraints and KL-based methods.

 **2. Expanded Explanations**:
- The revised manuscript includes thorough discussions on the metrics and methodology, highlighting their importance and relevance to the study’s objectives.
- We elaborated on how minimizing KL divergence margin balances alignment and diversity in model outputs, ensuring robust and controllable behavior.

 **3. Addressed Reviewers’ Concerns**:
- Additional context and justifications have been added for the use of feature-level constraints, specifically how they align with and complement KL-based approaches while improving computational efficiency.

We hope these revisions address your comments and concerns effectively. Please feel free to reach out if you have further questions or require additional clarifications. Your feedback has been instrumental in improving the quality and clarity of our work, and we thank you again for your constructive input.

---

### Meta-Review · Area_Chair_GVzr · 2024-12-21

**Metareview:**

This paper introduces **Feature-level Constrained Preference Optimization (FPO)** for aligning large language models (LLMs) with human preferences. The key contribution involves replacing token-level KL divergence constraints, as used in TDPO, with feature-level constraints derived from Sparse Autoencoders (SAEs). Additionally, the paper integrates a heuristic length control mechanism for response generation. The proposed approach demonstrates improved computational efficiency, evidenced by reductions in GPU memory usage, and marginal gains in evaluation metrics.

The primary novelty lies in leveraging sparse autoencoders to enable feature-level regularization, presenting it as a potentially more interpretable and computationally efficient alternative to token-level alignment techniques. This design choice highlights a unique perspective on preference optimization by reducing computational overhead while maintaining alignment stability.

However, the work has notable weaknesses. The approach combines elements of existing methods (e.g., DPO, SimPO, and TDPO) without introducing significant innovation. The substitution of token-level KL divergence with feature-level MSE regularization lacks theoretical justification for being a superior mechanism. Furthermore, the reported performance gains are small, inconsistent across datasets, and lack robust statistical validation. The experiments are limited to Gemma models, with no evidence of generalizability to other architectures like LLaMA or GPT.

In summary, while the proposed method is computationally efficient and introduces a novel application of feature-level constraints, the marginal performance improvements, lack of theoretical grounding, and restricted experimental scope weaken its impact. These limitations make it challenging to justify acceptance at this stage.

**Additional Comments On Reviewer Discussion:**

The authors provided extensive responses, addressing reviewer concerns about reproducibility, generation diversity, experimental setup, and the interpretability of sparse autoencoders. While one reviewer acknowledged the effort and improvements, another highlighted that the observed gains were marginal and lacked compelling evidence of practical impact. Despite the revisions and clarifications, fundamental concerns about the method’s effectiveness, theoretical foundation, and generalizability remain unresolved.

---

### Decision · Program_Chairs · 2025-01-22

Reject